# Snf1/AMPK fine-tunes TORC1 signaling in response to glucose starvation

Marco Caligaris[1], Raffaele Nicastro[1]*, Zehan Hu[1], Farida Tripodi[2], Johannes Erwin Hummel[3], Benjamin Pillet[1], Marie-Anne Deprez[4], Joris Winderickx[4], Sabine Rospert[3,5], Paola Coccetti[2], Jörn Dengjel[1], Claudio De Virgilio[1]*

[1]Department of Biology, University of Fribourg, Fribourg, Switzerland; [2]Department of Biotechnology and Biosciences, University of Milano-Bicocca, Milano, Italy; [3]Institute of Biochemistry and Molecular Biology, Faculty of Medicine, University of Freiburg, Freiburg, Germany; [4]Functional Biology, KU Leuven, Heverlee, Belgium; [5]Signalling Research Centres BIOSS and CIBSS, University of Freiburg, Freiburg, Germany

**\*For correspondence:**
Raffaele.Nicastro2@unifr.ch (RN);
Claudio.DeVirgilio@unifr.ch
(CDV)

**Competing interest:** The authors declare that no competing interests exist.

**Abstract** The AMP-activated protein kinase (AMPK) and the target of rapamycin complex 1 (TORC1) are central kinase modules of two opposing signaling pathways that control eukaryotic cell growth and metabolism in response to the availability of energy and nutrients. Accordingly, energy depletion activates AMPK to inhibit growth, while nutrients and high energy levels activate TORC1 to promote growth. Both in mammals and lower eukaryotes such as yeast, the AMPK and TORC1 pathways are wired to each other at different levels, which ensures homeostatic control of growth and metabolism. In this context, a previous study (Hughes Hallett et al., 2015) reported that AMPK in yeast, that is Snf1, prevents the transient TORC1 reactivation during the early phase following acute glucose starvation, but the underlying mechanism has remained elusive. Using a combination of unbiased mass spectrometry (MS)-based phosphoproteomics, genetic, biochemical, and physiological experiments, we show here that Snf1 temporally maintains TORC1 inactive in glucose-starved cells primarily through the TORC1-regulatory protein Pib2. Our data, therefore, extend the function of Pib2 to a hub that integrates both glucose and, as reported earlier, glutamine signals to control TORC1. We further demonstrate that Snf1 phosphorylates the TORC1 effector kinase Sch9 within its N-terminal region and thereby antagonizes the phosphorylation of a C-terminal TORC1-target residue within Sch9 itself that is critical for its activity. The consequences of Snf1-mediated phosphorylation of Pib2 and Sch9 are physiologically additive and sufficient to explain the role of Snf1 in short-term inhibition of TORC1 in acutely glucose-starved cells.

## Editor's evaluation

This rigorous and careful study provides some of the first mechanistic insights into the way that glucose starvation triggers inhibition of TORC1 (particularly in yeast) and will serve as an important resource for those interested in AMPK/Snf1 dependent regulation of a variety of other pathways and processes. The paper also provides the clearest picture yet of the regulation of Pib2, an important but poorly understood TORC1 regulator in yeast and likely beyond. The proposed mechanism is interesting and proposes multiple ways of interaction between the two signaling cascades, and will be of interest to researchers working on mechanisms of gene regulation by signaling pathways.

## Introduction

The eukaryotic target of rapamycin complex 1 (TORC1/mTORC1) signaling pathway serves as a central hub that couples growth signals with metabolic circuits that define cell growth. The TORC1 protein kinase activity is positively regulated by intracellular nutrients (*e.g.*, amino acids, glucose, and lipids), high energy levels, as well as extracellular growth factors (e.g. insulin and insulin-like growth factor 1 [IGF-1]). In response to these cues, it favors the increase of cellular mass by stimulating lipid, nucleotide, and protein synthesis and by inhibiting the autophagic recycling of macromolecules (*Albert and Hall, 2015*; *González and Hall, 2017*; *Laplante and Sabatini, 2012*; *Liu and Sabatini, 2020*). TORC1 function is embedded in as yet incompletely understood feedback loops, allowing it to act as a metabolic rheostat. Uncoupling TORC1 from this regulatory network is associated with dramatically altered lifespan in unicellular organisms such as yeast and with diseases such as cancer, immunodeficiency, type 2 diabetes, and neurodegeneration in humans (*Laplante and Sabatini, 2012*).

The core structure of TORC1 is highly conserved among eukaryotes and consists of a dimer of a heterotrimeric complex that harbors a TOR serine/threonine protein kinase (Tor1 or Tor2 in the budding yeast *Saccharomyces cerevisiae* or mTOR in mammals) and two regulatory proteins (yeast Kog1 and Lst8, or the orthologous mammalian Raptor [regulatory-associated protein of mTOR] and LST8 [mLST8], respectively *Wullschleger et al., 2006*). Additional non-conserved proteins, such as Tco89 in yeast or the proline-rich Akt substrate of 40 kDa (PRAS40) and the DEP domain-containing mTOR- interacting protein (DEPTOR) in mammals, associate with this core complex to adapt its function to species-specific requirements (*Loewith et al., 2002*; *Peterson et al., 2009*; *Reinke et al., 2004*; *Sancak et al., 2007*). TORC1 mainly functions at the vacuolar/lysosomal surface both in lower eukaryotes like the yeast *S. cerevisiae* as well as in higher eukaryotes such as *Drosophila* and mammals. At this location, TORC1 binds to and/or is regulated by the conserved heterodimeric Rag GTPases (i.e. yeast Gtr1 bound to Gtr2, or mammalian RagA or B bound to RagC or D). These heterodimers associate with structurally conserved protein complexes coined the EGO (exit from rapamycin-induced growth arrest) ternary complex in yeast (EGO-TC; comprising Ego1/Mhe1, Ego2, and Ego3/Slm4) or the pentameric Ragulator complex in mammals (comprising p18, p14, MP1, C7orf59, and HBXIP) (*Bar-Peled et al., 2012*; *Dubouloz et al., 2005*; *Powis et al., 2015*; *Sancak et al., 2010*). The complexes are anchored to vacuolar/lysosomal membranes through N-terminally lipidated Ego1 or p18, respectively (*Binda et al., 2009*; *Nada et al., 2009*; *Powis et al., 2015*; *Sancak et al., 2010*). The Rag GTPases adopt one of two stable conformations, an active state in which Gtr1 or RagA/B is bound to GTP and Gtr2 or RagC/D to GDP, and an inactive state with the opposite GTP/GDP-loading configuration. The respective nucleotide-loading states are primarily preserved by crosstalk between the Rag GTPases and are regulated via a set of conserved GTPase activating (GAP) protein complexes (i.e. yeast SEACIT/mammalian GATOR1 and yeast Lst4-Lst7/mammalian FNIP-FLCN acting on Gtr1/RagA/B and Gtr2/RagC/D, respectively *Bar-Peled et al., 2013*; *Panchaud et al., 2013a*; *Panchaud et al., 2013b*; *Péli-Gulli et al., 2015*; *Petit et al., 2013*; *Shen et al., 2017*; *Tsun et al., 2013*), which mediate cytosolic and/or vacuolar/lysosomal amino acid levels through different mechanisms (*González and Hall, 2017*; *Liu and Sabatini, 2020*; *Nicastro et al., 2017*). Notably, in flies and mammals, but likely not in yeast (*Powis and De Virgilio, 2016*), the Rag-GTPase tethered, lysosome-associated TORC1 pool is also allosterically activated by the small GTPase Rheb (Ras homolog enriched in brain) in its GTP-bound form (*Anandapadamanaban et al., 2019*; *Buerger et al., 2006*; *Long et al., 2005*; *Rogala et al., 2019*). Rheb responds, among other factors, to energy levels that are mainly integrated by the Rheb GAP complex (comprising TSC1, TSC2, and TBC1D7), which is also known as Rhebulator (*Demetriades et al., 2014*; *Dibble et al., 2012*; *Inoki et al., 2003a*; *Long et al., 2005*; *Tee et al., 2003*; *Yang et al., 2017*). This mechanism relies on the AMP-activated protein kinase (AMPK), a key regulator of cellular energy charge that inactivates mTORC1 indirectly by phosphorylating TSC2 and thereby activating the GAP activity of Rhebulator (*Inoki et al., 2003b*; *Shaw et al., 2004*).

AMPK functions within a conserved heterotrimeric complex encompassing a catalytic α subunit (yeast Snf1 or mammalian α1/2), and one of each β- (yeast Gal83, Sip1, and Sip2 or mammalian β1/2) and γ- (yeast Snf4 or mammalian γ1/2/3) regulatory subunits (*Carling, 2004*; *Ghillebert et al., 2011*; *Hedbacker and Carlson, 2008*). In line with its denomination, mammalian AMPK is allosterically activated by AMP. This seems not to be the case for Snf1, since the latter is mainly activated by the absence of glucose in the medium and since regulatory sites have been characterized that preferably bind ADP instead of AMP (*Coccetti et al., 2018*; *Herzig and Shaw, 2018*; *Mayer et al.,*

2011; *Wilson et al., 1996*). Nonetheless, both mammalian AMPK and yeast Snf1 are similarly activated by phosphorylation of a conserved threonine (Thr) within their activation loop (yeast Thr[210] or mammalian Thr[172]) that is executed by specific upstream kinases (yeast Sak1, Tos3, and Elm1 or mammalian LKB1 and CaMKK2) (*González et al., 2020*; *Hedbacker and Carlson, 2008*). In yeast, this phosphorylation event is reversed by the type I protein phosphatase Glc7 (combined with the regulatory subunit Reg1) specifically when extracellular glucose is sufficiently available (*Ludin et al., 1998*; *McCartney and Schmidt, 2001*). Both AMPK and Snf1 are also wired to control mTORC1/TORC1 via Raptor and Kog1, respectively, albeit through different mechanisms. Accordingly, AMPK has been reported to inhibit mTORC1 through direct phosphorylation of Raptor (*Gwinn et al., 2008*). Although one of the respective AMPK target residues in Raptor is conserved in Kog1, Snf1 does not control TORC1 through this residue (Ser[959]; *Kawai et al., 2011*), but rather through phosphorylation of Ser[491] and Ser[494], thereby promoting the formation of TORC1-bodies during prolonged glucose starvation (*Hughes Hallett et al., 2015*; *Sullivan et al., 2019*). Notably, the latter process is also regulated by Pib2, a phosphatidylinositol-3-phosphate (PI3P) and Kog1-binding protein that senses glutamine levels and that can both activate and inhibit TORC1 through its C-terminal and N-terminal domains, respectively (*Hatakeyama, 2021*; *Kim and Cunningham, 2015*; *Michel et al., 2017*; *Sullivan et al., 2019*; *Tanigawa and Maeda, 2017*; *Tanigawa et al., 2021*; *Troutman et al., 2022*; *Ukai et al., 2018*; *Varlakhanova et al., 2017*). Finally, Snf1 also plays a role in maintaining TORC1 inactive during the early phase following acute glucose starvation, which is independent of TORC1-body formation, but the underlying mechanism is still elusive (*Hughes Hallett et al., 2015*).

To address the outstanding question of how Snf1 contributes to TORC1 inhibition following glucose starvation, we used a yeast strain in which Snf1 can be conditionally inactivated by addition of the ATP-analog 2NM-PP1 and applied a mass spectrometry (MS)-based phosphoproteomics strategy that combines in vivo proteomics with on-beads in vitro kinase assays (OBIKA) to identify direct Snf1 target residues on a global scale. This approach not only allowed us to uncover the currently largest set of Snf1-dependent phosphorylation events in *S. cerevisiae*, but also pinpointed several potential Snf1 targets within the TORC1 signaling pathway. Employing genetic, biochemical, and physiological experiments, we demonstrate that Snf1 temporally maintains TORC1 inactive in glucose-starved cells primarily through the regulatory protein Pib2. In addition, Snf1 specifically phosphorylates the TORC1 effector kinase Sch9 and thereby antagonizes the phosphorylation of a C-terminal TORC1-target residue within Sch9 that is critical for its activity. The consequences of Snf1-mediated phosphorylation of Pib2 and Sch9 are physiologically additive and sufficient to mediate an appropriate short-term response of TORC1 to acute glucose starvation.

## Results

### Snf1 prevents transient reactivation of TORC1 in glucose-starved cells

To study how Snf1 contributes to the inhibition of TORC1 in glucose-starved cells, we used a strain in which the *SNF1* locus expresses the *snf1[I132G]*-allele (Snf1[as]) that is sensitive to the ATP-analog 2-naphthylmethyl pyrazolopyrimidine 1 (2NM-PP1) and that supports normal growth on sucrose or low glucose-containing media (*Shirra et al., 2008*; *Zaman et al., 2009*). In our control experiments, the Snf1[as] allele was appropriately activated by its upstream protein kinases, as assessed by the rapid increase in Thr[210] phosphorylation that was comparable to wild-type Snf1, and it mediated the rapid phosphorylation of a synthetic AMPK activity reporter substrate (ACC1-GFP) in glucose-starved cells, albeit to a significantly lower extent than wild-type Snf1 (*Figure 1A and B*, and *Figure 1—figure supplement 1A*). Notably, nitrogen starvation also activates Snf1, but much less than glucose starvation (*Figure 1—figure supplement 1B, C*). As expected, the presence of 2NM-PP1 fully abrogated the activity of Snf1[as], but not that of wild-type Snf1 (*Figure 1—figure supplement 1A*), while its association with the Snf1[as] kinase active site protected Snf1[as]-pThr[210] from dephosphorylation as reported earlier (*Chandrashekarappa et al., 2013*). Compared to DMSO-treated (control) *snf1[as]* cells, 2NM-PP1-treated *snf1[as]* cells were also significantly compromised in maintaining TORC1 inactive, as detected by measuring the phosphorylation of Thr[737] in Sch9, a proxy for TORC1 activity (*Urban et al., 2007*), specifically during the time frame of 6–15 min following glucose starvation (*Figure 1A–D*). Because loss of Snf1 caused a comparable defect that was independent of the presence or absence of 2NM-PP1 (*Figure 1A*), our data corroborate the previous notion that Snf1 activity is required for

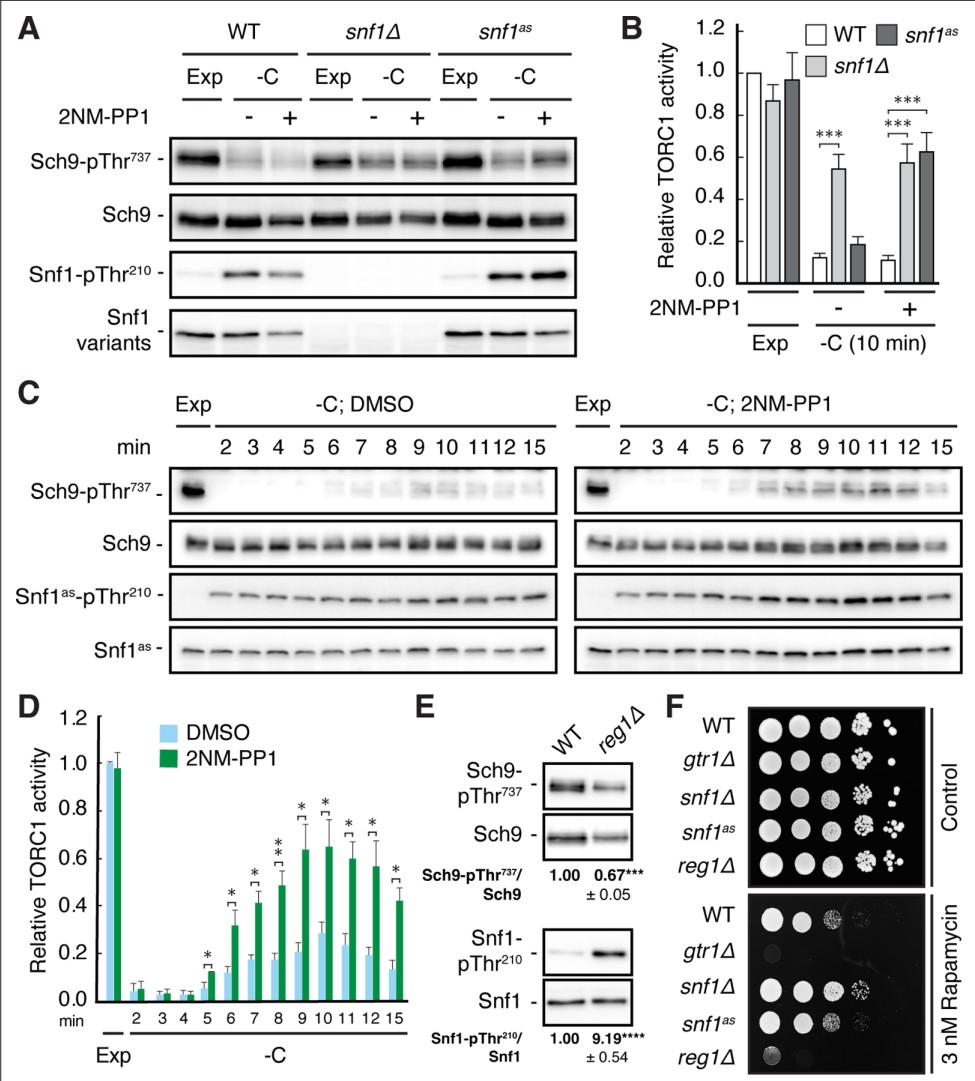

**Figure 1.** Snf1 is required for proper downregulation of TORC1 in glucose-starved cells. (**A, B**) Wild-type (WT), *snf1Δ*, and *snf1as* (analog-sensitive) cells were grown exponentially (Exp) and starved for glucose for 10 min (-C) in the absence (-; DMSO vehicle control) or presence (+) of 2NM-PP1. Phosphorylation of the *bona fide* TORC1 target residue Thr737 in Sch9 and of Thr210 in Snf1 was detected by immunoblot analyses of whole cell extracts using phospho-specific antibodies against the respective phospho-residues. Anti-Sch9 and anti-His6 antibodies served to detect the levels of Sch9 and Snf1, respectively (**A**). Notably, binding of 2NM-PP1 to the catalytic cleft of Snf1as inhibits its kinase activity and, at the same time, prevents the dephosphorylation of phosphorylated Thr210 (pThr210) in Snf1 (***Chandrashekarappa et al., 2013***). The mean TORC1 activities (*i.e.* Sch9-pThr737/Sch9) were quantified, normalized relative to exponentially growing WT cells (set to 1.0), and shown in the bar diagram in (**B**) (n=6; + SEM; unpaired Student's t-test, ***≤0.0005). (**C, D**) Analog-sensitive *snf1as* cells were treated as in (**A**), but harvested at the times indicated following glucose starvation (-C). The respective relative TORC1 activities were assessed as in (**B**), but cross-normalized (for each of the two sets of blots) to the same sample from exponentially growing cells (lane 1; Exp), and are shown in (**D**) (n=4; + SEM; unpaired Student's t-test, *p≤0.05, **p≤0.005). (**E**) WT and *reg1Δ* cells were grown exponentially and assayed for their mean relative TORC1 activities (Sch9-pThr737/Sch9) and Snf1-Thr210 phosphorylation levels (Snf1-pThr210/Snf1), each normalized to WT cells (set to 1.0; n=4; ± SEM). In unpaired Student's tests, both values in *reg1Δ* cells were significantly different from the ones in WT cells (***p≤0.0005, ****p≤0.00005). (**F**) Exponentially growing cells (of the indicated genotype) were 10-fold serially diluted, spotted on synthetic complete medium containing, or not (control), 3 nM rapamycin, and grown for 3 days at 30 °C (n=3). The online version of this article includes the following source data for *Figure 1—source data 1*, quantification of blots for graphs shown in (**B, D and E**); *Figure 1—source data 2*, uncropped blots shown in (**A, C and E**); Uncropped dropspots shown in (**F**); *Figure 1—source data 3*, raw blots shown in (**A, C and E**) and raw dropspots shown in (**F**).

*Figure 1 continued on next page*

*Figure 1 continued*

The online version of this article includes the following source data and figure supplement(s) for figure 1:

**Source data 1.** Quantification of blots for graphs shown in B, D, E.

**Source data 2.** Uncropped blots shown in A, C, E and uncropped dropspots shown in F.

**Source data 3.** Raw blots shown in A, C, E and raw dropspots shown in F.

**Figure supplement 1.** Specificity of 2NM-PP1 and differential Snf1 activation upon nitrogen and carbon starvation.

**Figure supplement 1—source data 1.** Quantification of blots for graphs shown in C.

**Figure supplement 1—source data 2.** Uncropped blots shown in A, B.

**Figure supplement 1—source data 3.** Raw blots shown in A, B.

maintaining TORC1 inactive specifically during the early, but not at later, phases of glucose starvation (*Hughes Hallett et al., 2015*). Based on these observations, we reasoned that unscheduled hyperactivation of Snf1 might also inhibit TORC1 even in cells growing in a glucose-rich environment. This was indeed the case as loss of Reg1, the regulatory subunit that instructs the PP1 Glc7 to dephosphorylate pThr$^{210}$ and thus inactivate Snf1 (*Ludin et al., 1998*; *Sanz et al., 2000*), resulted in hyperphosphorylation of Snf1-Thr$^{210}$ that was accompanied by a significant reduction in TORC1 activity (*Figure 1E*). In line with these data, loss of Reg1, like loss of the TORC1-regulatory Rag GTPase Gtr1, rendered cells sensitive to sub-inhibitory levels of rapamycin (as also observed earlier; *Bertram et al., 2002*), while loss of Snf1 conferred slight resistance to rapamycin (*Figure 1F*).

It has previously been demonstrated that TORC1 activity can be transiently (on a short-time scale of 1–5 min) activated by the addition of a nitrogen source such as glutamine to nitrogen starved cells (*Stracka et al., 2014*). Based on our results above, we therefore assumed that the respective transient activation of TORC1 may be reduced in the absence of glucose, because we expected this to activate Snf1 and therefore (directly or indirectly) inhibit TORC1. Indeed, when *snf1$^{as}$* cells were starved for both glucose and glutamine at the same time, TORC1 was rapidly inactivated and could only be transiently reactivated when cells were refed with either glucose alone or with glucose and glutamine combined (*Figure 2A–D*), but not when refed with glutamine alone (*Figure 2E and F*). In the latter case, however, 2NM-PP1 treatment, and hence inactivation of Snf1$^{as}$, was able to partially restore the transient glutamine-mediated TORC1 activation (*Figure 2E and F*). From these experiments, we infer that Snf1 not only contributes to proper TORC1 inhibition in glucose-starved cells, but also helps to prevent the transient reactivation of TORC1 in glutamine refed, glucose-starved cells.

## Global phosphoproteomics identifies potential Snf1 targets within the TORC1 signaling pathway

To dissect the mechanisms by which Snf1 impinges on TORC1 signaling, we decided to perform a set of stable isotope-labeling by amino acids in cell culture (SILAC)-based quantitative phosphoproteomic experiments, similarly as recently described (*Hu et al., 2021*). For in vivo analyses, *snf1$^{as}$* cells were grown in three different SILAC media supporting the comparison of three experimental conditions (*Figure 3A*). Cells grown in full medium (2% glucose, light label) served as control. The cellular response to glucose starvation was analyzed after 5 and 15 min (shift to 0.05% glucose, medium and heavy label, respectively). To discriminate potential Snf1 target sites from sites being regulated by other kinases, concomitantly to the glucose downshift, cells were treated, or not, with the selective ATP-competitive inhibitor 2NM-PP1. Snf1 target sites should be positively regulated in the control set and not, or significantly less, in the 2NM-PP1-treated set (*Figure 3A*; n=5 biological replicates per set). To further discriminate direct from indirect effects, we performed whole proteome on bead in vitro kinase assays (OBIKA) comparing the immobilized proteome of *snf1$^{as}$* cells (growing exponentially in the presence of 5% glucose and ATP analog) incubated with purified wild-type Snf1 to that incubated with a kinase-inactive Snf1$^{T210A}$ mutant (*Figure 3A*; n=5 biological replicates) (*Hu et al., 2021*).

In vivo analyses led to the identification of 40'547 phosphosites, of which 34'747 could be quantified (*Figure 3B*). After stringent filtering, we used 21'223 sites, which clearly localized to a specific amino acid residue (localization probability >0.75) (*Olsen et al., 2006*; *Supplementary file 2A-B*), for statistical analyses to identify potential Snf1 target sites. Class I target sites had to fulfill two criteria: sites had to be significantly upregulated (i) when using a random effect model comparing starved

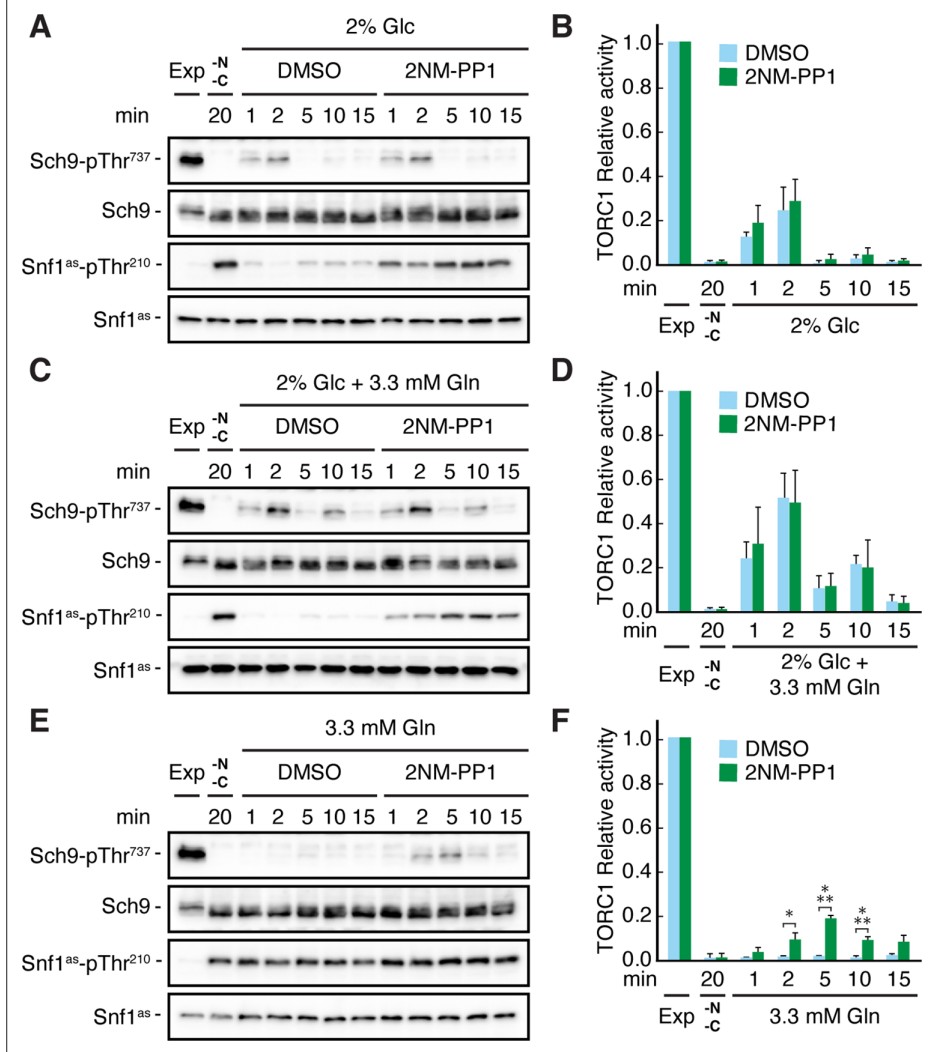

**Figure 2.** Snf1 prevents transient TORC1 restimulation by glutamine in glucose- and glutamine-starved cells. (**A–F**) Exponentially growing *snf1as* cells were starved for 20 min for nitrogen and glucose (-N, -C) and then restimulated for the times indicated, in the absence (DMSO) or presence of 2NM-PP1, with 2% glucose (Glc) (**A, B**), 2% glucose and 3.3 mM glutamine (Gln) (**C, D**), or 3.3 mM glutamine (**E, F**). Immunoblot analyses were performed as in *Figure 1A* and the relative TORC1 activities in (**A**), (**C**), and (**E**), were quantified as in *Figure 1B* and are shown in (**B**), (**D**), and (**F**), respectively (n=3; + SEM; unpaired Student's t-test, *p≤0.05, ***p≤0.0005). The online version of this article includes the following source data for *Figure 2—source data 1*, quantification of blots for graphs shown in (**B, D and F**); *Figure 2—source data 2*, uncropped blots shown in (**A, C and E**); *Figure 2—source data 3*, raw blots shown in (**A, C and E**).

The online version of this article includes the following source data for figure 2:

**Source data 1.** Quantification of blots for graphs shown in B, D and F.

**Source data 2.** Uncropped blots shown in A, C and E.

**Source data 3.** Raw blots shown in A, C and E.

(5 min and 15 min) to non-starved cells (p<0.05); and (ii) when comparing starved cells to cells treated additionally with the inhibitor 2NM-PP1 (Student's t-test, FDR<0.05). Class II target sites had only to fulfill the second criterium. Class II sites contained target residues such as Ser[1157] in Acc1, the cyto-solic acetyl-CoA carboxylase that is already phosphorylated by Snf1 in cells grown on 2% glucose, but dephosphorylated upon 2NM-PP1-mediated inhibition of Snf1as (*Braun et al., 2014*). In total, this led to a shortlist of 1409 sites, divided into 984 class I sites and 425 class II sites (*Figure 3B* and *Supplementary file 2C*). Our shortlisted class I and class II sites cover between 26% and 53% of the

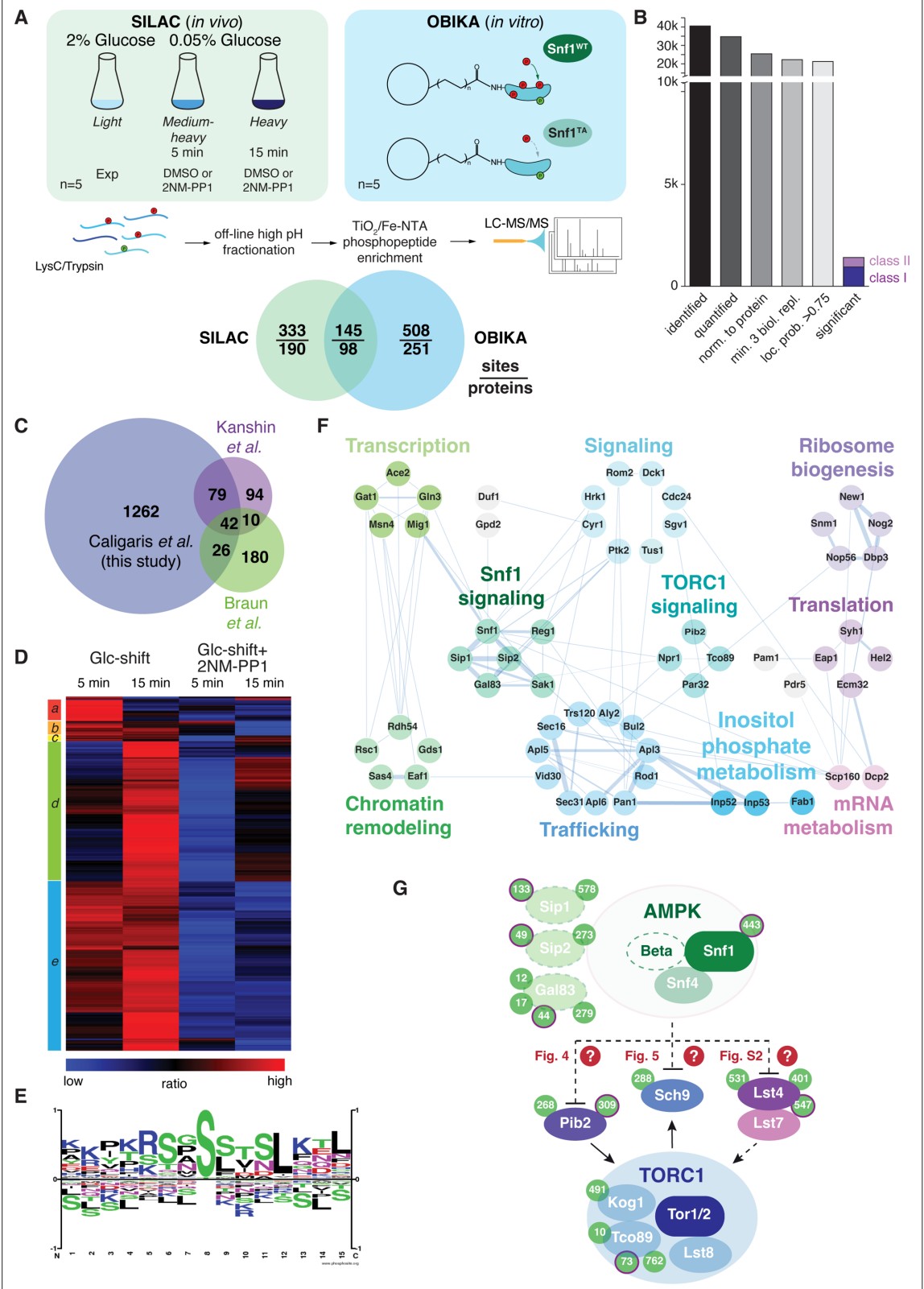

**Figure 3.** Quantitative phosphoproteomic analyses for the identification of potential Snf1 target sites. (**A**) Quantitative MS-based proteomics workflow (SILAC n=5, OBIKA n=5). (**B**) Histogram of the number of identified phosphosites in the SILAC analysis after each filtering step and the relative amount of phosphosites attributed to classes I and II among the significant ones. (**C**) Proportional Venn diagram highlighting the commonly identified phosphosites in the current and two recent Snf1 phosphoproteomic studies. (**D**) Heatmap of phosphosite kinetics. Normalized SILAC ratios (treated

*Figure 3 continued on next page*

Figure 3 continued

*versus* untreated) of class I and class II sites (highlighted in B) were log2 transformed and z normalized prior unsupervised hierarchical clustering of the rows tree using Euclidean distance as matrix. Five major clusters *a-e*, each highlighted by a different color, are observed. (**E**) Motif analyses of Snf1 phosphosites identified by in vivo SILAC and OBIKA experiments as outlined in (**A**). (**F**) Protein-protein interaction network comprising 57 interconnected proteins out of the 98 where at least one Snf1-dependent phosphosite was shortlisted by the intersection of the SILAC and OBIKA analyses. The network was generated with Cytoscape using the STRING plugin, setting the confidence (score) cut-off at 0.25. Edge thickness represents the score value of each interaction. (**G**) Schematic model representing components of the Snf1/AMPK and TORC1 signaling pathways that contain Snf1-regulated phosphosites. Phosphosites attributed to class II are highlighted with a bold-pink outline. The three Snf1 β-subunits are represented with a dashed outline on the left. Solid and dashed arrows refer to direct and indirect activating interactions, respectively. Dashed lines with bars refer to potential (question marks) inhibitory interactions that are experimentally addressed in the indicated figures in this study. Source data for this figure are provided in *Supplementary file 2*.

The online version of this article includes the following source data and figure supplement(s) for figure 3:

**Figure supplement 1.** Specific motif analyses of Snf1 phosphosites and role of Lst4 in TORC1 reactivation in glucose-starved, Snf1-inhibited cells.

**Figure supplement 1—source data 1.** Quantification of blots for values shown in C.

**Figure supplement 1—source data 2.** Uncropped blots shown in C.

**Figure supplement 1—source data 3.** Raw blots shown in C.

previously published potential Snf1 target sites (*Braun et al., 2014*; *Kanshin et al., 2017*; *Supplementary file 2D*), overlapping to a larger extent with the respective studies than the studies with each other (*Figure 3*). In addition, we expand the potential Snf1 target repertoire by more than 1200 sites highlighting the resource character of the current study. We performed hierarchically clustering to characterize the kinetic behavior of sites and observed five major clusters (*Figure 3D*): cluster *a* contains sites that responded transiently after 5 min of stimulation; clusters *b*, *c*, and *e* contain sites that responded in a sustained manner after 5 min and 15 min; and cluster *d* contains responders that reacted only after 15 min. Source proteins of cluster *a* relate to 'regulation of growth', amongst them the Snf1 target Mig1, while those in clusters *b*, *c*, and *e* are linked to 'cell cycle, transcription, endocytosis, and proteins transport', such as the zinc-finger and Snf1 target protein Msn4, and those in cluster *d* are enriched in 'serine-threonine kinases', like Atg1 and Sch9 (*Supplementary file 2E*).

To discriminate potential indirect effects observed in vivo from direct Snf1 targets, we overlaid the significantly regulated sites from the SILAC and OBIKA datasets. Of the 986 commonly identified and quantified sites, 145 sites on 98 source proteins were significantly regulated by both approaches, characterizing them as *bona fide* Snf1 target sites (*Figure 3A* and *Supplementary file 2F*). The consensus motif of these sites corroborates the published AMPK and Snf1 motif with two basic amino acid residues in the –3 and –4 positions and a hydrophobic leucine residue in the +4 position (*Figure 3E*; *Dale et al., 1995*; *Gwinn et al., 2008*; *Schaffer et al., 2015*; *Scott et al., 2002*). As such, this further supports our interpretation that these sites are *bona fide* Snf1 target sites. Notably, of the 145 confirmed target sites, 81 (i.e. 72%) were significantly regulated after both 5 min and 15 min. Of the remaining 64 sites, 32 responded only after 5 min, while the other 32 responded only after 15 min. Some of the former residues are located within Snf1 itself, the β-subunit of the Snf1 complex (i.e. Sip1), the Snf1-targeting kinase Sak1, and Mig1, while some of the latter are located within the known Snf1-interacting proteins such as Gln3, Msn4, and Reg1. These observations indicate that Snf1-dependent phosphorylation initiates, as expected, within the Snf1 complex and then progresses to other effectors. Interestingly, based on the residues that responded exclusively after 5 min, we retrieved a perfect Snf1 consensus motif (i.e. an arginine residue in the –3 position and a leucine residue in the +4 position; *Figure 3—figure supplement 1A*). The one retrieved for the residues that respond exclusively at 15 min, in contrast, significantly deviated from this consensus motif (*Figure 3—figure supplement 1B*). The temporal deferral of some Snf1 target phosphorylation events may therefore perhaps be in part be explained by reduced substrate affinity due to consensus motif divergence.

As a kinase generally regulates multiple proteins within a given pathway, we analyzed known protein-protein interactions of the 98 Snf1 target proteins in STRING DB, enabling us to generate a network of 57 proteins (*Figure 3F*; *Szklarczyk et al., 2021*). In line with the significant overlap of our data with the ones published by *Braun et al., 2014*, this network covers similar biological processes including intracellular trafficking, ribosome biogenesis, translation, mRNA metabolism, inositol phosphate metabolism, chromatin remodeling, and TORC1 signaling (*Figure 3F*). Gratifyingly, this network also includes most of the previously known proximal Snf1 targets including for instance Ccr4

(*Braun et al., 2014*), Cyr1/Cdc35 (*Nicastro et al., 2015*), Eap1 (*Braun et al., 2014*). Gat1 (*Kulkarni et al., 2006*), Gln3 (*Bertram et al., 2002*; *Kulkarni et al., 2006*), Mig1 (*DeVit and Johnston, 1999*; *Ostling and Ronne, 1998*; *Smith et al., 1999*; *Treitel et al., 1998*), Msn4 (*Estruch and Carlson, 1993*; *Petrenko et al., 2013*), and Rod1 (*Alvaro et al., 2016*; *Laussel et al., 2022*; *O'Donnell and Schmidt, 2019*; *Shinoda and Kikuchi, 2007*). In addition, the dataset also pinpoints potential Snf1 target residues in numerous proteins that have an assigned function as Snf1 effectors such as Aly2 and Bul2 (*Bowman et al., 2022*; *O'Donnell and Schmidt, 2019*; *Ptacek et al., 2005*), Glo3 (*Arakel et al., 2019*), Gpd2 (*Lee et al., 2012*), and Npr1 (*Brito et al., 2019*). Finally, Snf1-regulated residues were also identified in the Snf1-β-subunits Gal83 and Sip2 (*Carling, 2004*; *Hedbacker and Carlson, 2008*), the Sak1 kinase that phosphorylates Snf1-Thr$^{210}$ (*González et al., 2020*; *Hedbacker and Carlson, 2008*), the PP1 phosphatase Glc7-regulatory subunit Reg1 and the protein phosphatase C Ptc1 that share an overlapping role in the dephosphorylation of Snf1-pThr$^{210}$ (*Ludin et al., 1998*; *McCartney and Schmidt, 2001*; *Ruiz et al., 2013*), and Snf1 itself. These observations indicate that Snf1 is embedded and engaged in an elaborated feedback control network, which, based on our current data, can be experimentally addressed in the future.

As briefly mentioned above, our in vivo phosphoproteomic analyses also pinpointed several Snf1-regulated residues within proteins that act in the TORC1 signaling pathway, including Lst4 and Pib2 that function upstream of TORC1, the TORC1 subunits Kog1 and Tco89, and the proximal TORC1 effector Sch9 (*Urban et al., 2007*; *Figure 3G*). From these 5 TORC1-related hits, we excluded Kog1 from further analyses because the potential Snf1 target residue in Kog1 that we identified was Ser$^{491}$, and mutation of this residue has previously been found not to affect the capacity of cells to maintain TORC1 inactive during the early phase of glucose starvation (*Hughes Hallett et al., 2015*; *Sullivan et al., 2019*). We have also not given priority to the analysis of Tco89, since it is a heavily phosphorylated protein (>70 phosphorylated residues) that we intend to dissect separately in parallel studies. Finally, we also did not follow up on the TORC1-stimulating protein Lst4. The reason for this is that loss of Lst4 did not significantly change the transient reactivation of TORC1 in glucose-starved, 2NM-PP1-treated *snf1$^{as}$* cells (*Figure 3—figure supplement 1C*), even though it reduced the TORC1 activity in exponentially growing cells as reported (*Péli-Gulli et al., 2015*). Thus, we focused our subsequent experiments on the two remaining proteins, namely Pib2 and Sch9, and examined in more detail the consequences of their phosphorylation by Snf1 for signaling via the TORC1 pathway.

## Snf1 phosphorylates Pib2-Ser$^{268,309}$ to weaken its association with Kog1

Our phosphoproteomic approach identified two potential Snf1 target residues each within the Kog1-binding region of Pib2, Ser$^{268}$ and Ser$^{309}$, of which the latter was also detected by OBIKA with Snf1 (*Figure 4A–C* and *Supplementary file 2C-F*). To further corroborate these data, we carried out an in vitro Snf1 kinase assay using an N-terminally truncated Pib2 fragment as substrate, which, unlike full-length Pib2, could be stably expressed in and purified from bacteria. Accordingly, wild-type Snf1 (purified from yeast), but not the kinase-inactive Snf1$^{T210A}$, was able to phosphorylate the Pib2 fragment. We also introduced mutations of Ser$^{268}$ and Ser$^{309}$ in the Pib2 fragment and replaced them with alanine (Ala) residues. The fragments carrying either one of these mutations were still phosphorylated by wild-type Snf1, though to a different extent, but when both mutations were combined, the phosphorylation level dropped significantly (*Figure 4D*). Together with our finding that the Snf1 complex was able to bind Pib2 in microscale thermophoresis assays (*Figure 4E*), our data therefore led us to infer that Snf1 may control Pib2 function primarily through phosphorylation of the Ser$^{268}$ and Ser$^{309}$ residues.

To address the physiological role of Ser$^{268/309}$ phosphorylation in Pib2, we next studied the response to short-term glucose starvation of 2NM-PP1-treated and -untreated *snf1$^{as}$* strains that expressed either the phosphomutant Pib2$^{S268A,S309A}$ or the phosphomimetic Pib2$^{S268E,S309E}$. The *snf1$^{as}$* strain in which *PIB2* was deleted served as control. Interestingly, expression of Pib2$^{S268E,S309E}$ or loss of Pib2, but not the expression of Pib2$^{S268A,S309A}$, rendered TORC1 more sensitive to glucose-starvation in DMSO-treated control cells and largely suppressed the unscheduled reactivation of TORC1 in glucose-starved cells in which Snf1$^{as}$ was inhibited by a 2NM-PP1-treatment (*Figure 4F and G*). The observation that loss of Pib2 and expression of the Pib2$^{S268E,S309E}$ allele similarly affect TORC1 activity suggests that the latter allele may be compromised in a TORC1 activating mechanism. The Snf1-dependent phosphorylation of Pib2 may therefore possibly compromise the function of the C-terminal TORC1-activating domain

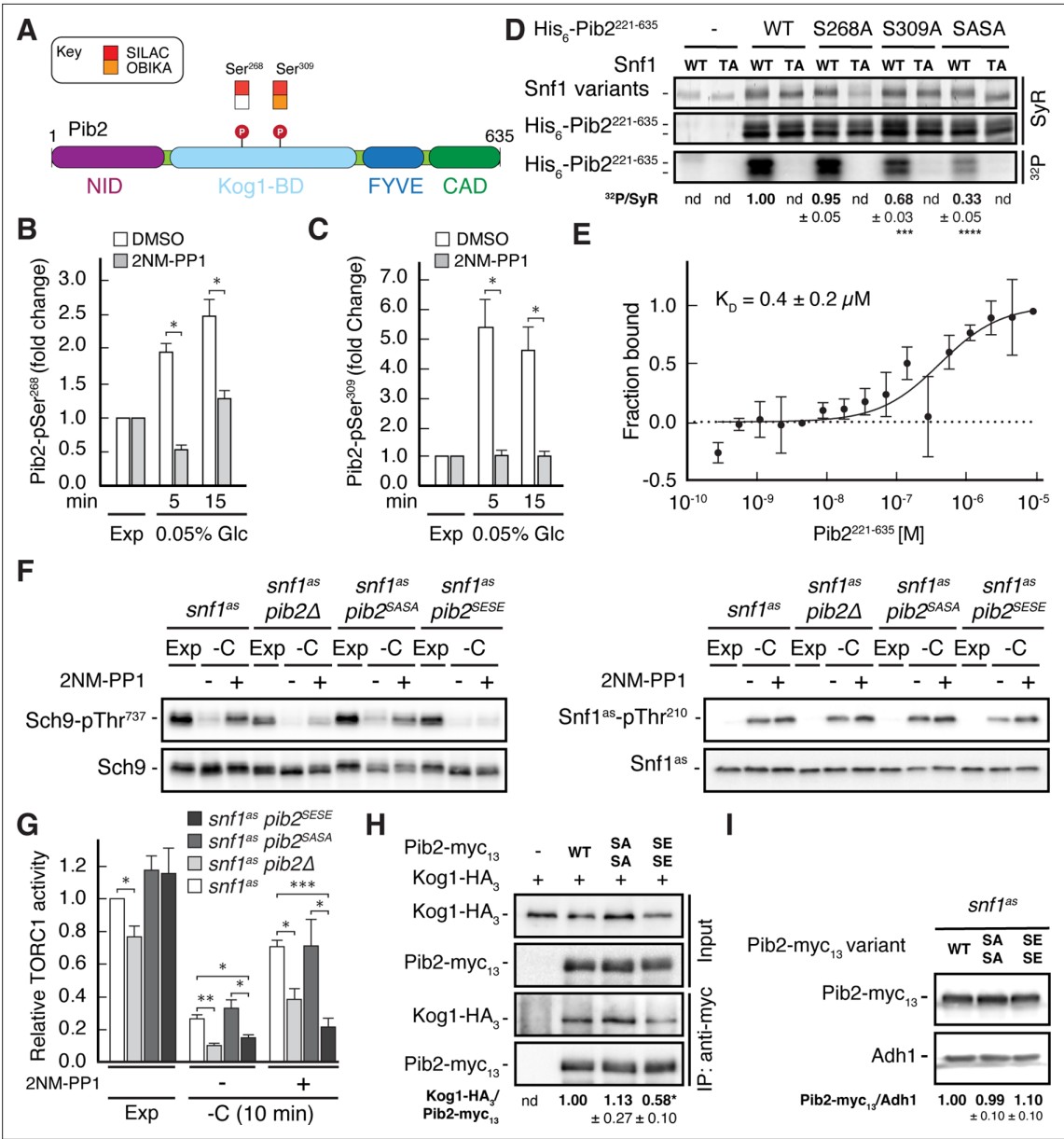

**Figure 4.** Snf1 weakens the Pib2-Kog1 association by phosphorylating Pib2-Ser[268,309.] (**A**) Schematic representation of the structure of Pib2 with the N-terminal TORC1-inhibitory (NID), Kog1-binding (Kog1-BD), phosphatidylinositol-3-phosphate (PI3P) -binding Fab1-YOTB-Vac1-EEA1 (FYVE), and C-terminal TORC1-activatory (CAD) domains (**Hatakeyama, 2021**). The residues Ser[268] and Ser[309] in Pib2 were both identified as potential Snf1 targets (P) in vivo (SILAC), while only Ser[309] was recovered in our highly multiplexed on-beads in vitro kinase assays (OBIKA) with Snf1. (**B, C**) Snf1 controls the phosphorylation of Ser[268] and Ser[309] in Pib2 in vivo. Phosphorylation levels of Pib2-Ser[268] (**B**) and Pib2-Ser[309] (**C**) in untreated (DMSO) and 2NM-PP1-treated $snf1^{as}$ cells that were grown exponentially (Exp) and limited for glucose (0.05%) for the times indicated. Mean values were extracted from the SILAC experiment (**Supplementary file 2** and **Figure 3**) and normalized to the ones in exponentially growing cells (set to 1.0) (n=3; + SD; unpaired Student's t-test, *FDR ≤ 0.05). (**D**) Snf1 phosphorylates Ser[268] and Ser[309] in Pib2 in vitro. Snf1 (WT) and kinase-inactive Snf1[T210A] (TA) were purified from yeast and used in protein kinase assays with [γ-[32]P]-ATP and a bacterially-expressed fragment of Pib2 (Pib2[221-635]) lacking the N-terminal 220 amino acids that, according to our in vivo proteomics analyses, did not contain a potential Snf1 target residue. In parallel protein kinase assays, we also used the respective Pib2[S268A], Pib2[S309A], and Pib2[S268A/S309A] (Pib2[SASA]) mutant fragments as substrates. Substrate phosphorylation was detected by autoradiography ([32]P, lower panel) and Sypro Ruby (SyR) staining is shown as loading control for the Snf1 variants (upper panel) and the His[6]-tagged Pib2 fragments that were partially degraded and ran in more than 1 band (panel in the middle). The mean phosphorylation of Pib2[S268A], Pib2[S309A], and Pib2[S268A/S309A] (Pib2[SASA]) fragments by wild-type Snf1 (i.e. [32]P signal/Sypro Ruby [SyR] substrate input level [including the indicated faster migrating proteolytic forms]; nd, not detected) was assessed relative to the one of the Pib2[WT] fragment (set to 1.0; n=3; ± SEM; unpaired Student's t-test, ***p≤0.0005, ****p≤0.00005). (**E**) The Snf1 complex binds Pib2. The binding affinity between bacterially purified, titrated Pib2[221-635] and yeast purified Snf1 complex (containing the C-terminally GFP-tagged Snf4 γ-subunit) was assessed by microscale thermophoresis. The dissociation constant (K[D]; 95% profile likelihood = 136–1098 nM,

*Figure 4 continued on next page*

*Figure 4 continued*

n=3; ± SEM) was calculated using a nonlinear asymmetric sigmoidal regression. (**F, G**) Expression of the phosphomimetic Pib2$^{S268E,S309E}$ allele, like loss of Pib2, rescues the TORC1 inactivation defect in glucose-starved, Snf1-compromised cells. Exponentially growing *snf1$^{as}$*, *snf1$^{as}$ pib2Δ*, *snf1$^{as}$ pib2$^{S268A,S309A}$* (*pib2$^{SASA}$*), and *snf1$^{as}$ pib2$^{S268E,S309E}$* (*pib2$^{SESE}$*) cells were grown exponentially (Exp) and then starved for 10 min for glucose (-C) in the absence (-; DMSO) or the presence (+) of 2NM-PP1. Immunoblot analyses of Sch9-pThr$^{737}$ and Sch9 (left blots) and of Snf1$^{as}$-pThr$^{210}$ and Snf1$^{as}$ (right blots) were carried out as in *Figure 1A* (**F**). The mean relative TORC1 activities in the four strains were quantified as in *Figure 1B* and normalized to the values in exponentially growing *snf1$^{as}$* cells (set to 1.0; n=4; + SEM; unpaired Student's t-test, *p≤0.05, **p≤0.005, ***p≤0.0005) (**G**). (**H**) The phosphomimetic Pib2$^{S268E,S309E}$ is compromised for TORC1-binding. Kog1-HA$_3$-expressing cells co-expressing Pib2-myc$_{13}$ (WT), Pib2$^{S368A,S309A}$-myc$_{13}$ (SASA), Pib2$^{S368E,S309E}$-myc$_{13}$ (SESE), or untagged Pib2 (-) were grown exponentially. Lysates (input) containing 60 mM glutamine and anti-myc immunoprecipitates (IP: anti-myc) were analyzed by immunoblotting with anti-HA and anti-myc antibodies, respectively. The mean relative amount of Kog1-HA$_3$ that was immunoprecipitated with Pib2-myc$_{13}$ and its variants was determined and normalized to the one between Kog1-HA$_3$ and Pib2-myc$_{13}$ (set to 1.0; n=4; + SEM; unpaired Student's t-test, *p≤0.05; nd, not detected). (**I**) Pib2-myc$_{13}$ alleles are adequately expressed. Expression of Pib2-myc$_{13}$ variants (as in F) was probed by immunoblot analysis in extracts of exponentially growing cells using anti-myc antibodies. Values were quantified relative to Adh1 levels (detected with anti-Adh1 antibodies) and normalized to the respective Pib2-myc$_{13}$/Adh1 ratio in cells expressing the WT Pib2-myc$_{13}$ (n=3; ± SEM; unpaired Student's t-test; nd, not detected). The online version of this article includes the following source data for *Figure 4—source data 1*, data for the graph shown in (**B, C and E**) and quantifications of the blots in (**D, H and I**) and for the graph shown in (**G**); *Figure 4—source data 2*, uncropped blots, gels and autoradiographies shown in (**D, F, H and I**); *Figure 4—source data 3*, raw blots, gels and autoradiographies shown in (**D, F, H and I**).

The online version of this article includes the following source data for figure 4:

**Source data 1.** Data for the graph shown in B, C and E and quantifications of the blots in D, H and I and for the graph shown in G.

**Source data 2.** Uncropped blots, gels and autoradiographies shown in D, F, H and I.

**Source data 3.** Raw blots, gels and autoradiographies shown in D, F, H and I.

(CAD) rather than the N-terminal TORC1-inhibitory domain (NID). Consequently, loss of Snf1, would allow the Pib2-CAD to transiently activate TORC1 in glucose-starved cells.

Notably, as a readout for TORC1 activity, we monitored the phosphorylation of the downstream effector kinase Sch9, which as mentioned above and further explained below, is itself a potential direct Snf1 substrate. Therefore, we sought to understand how Pib2 phosphorylation may affect TORC1 signaling. We noticed that the Ser$^{268}$ and Ser$^{309}$ Snf1 target sites reside within the Kog1-binding domains of Pib2 (*Michel et al., 2017*; *Sullivan et al., 2019*; *Troutman et al., 2022*), which led us to speculate that the phosphorylation state of these residues could affect the Pib2-Kog1 association. To this end, we performed co-immunoprecipitation assays and found that this was indeed the case, as we observed that the Pib2$^{S268E,S309E}$ variant exhibited significantly reduced affinity for Kog1 when compared to wild-type Pib2 or Pib2$^{S268A,S309A}$ (*Figure 4H*). Importantly, immunoblot analyses of the myc-tagged phosphomutant and phosphomimetic Pib2 variants showed that the introduction of the respective mutations in Pib2 did not affect the stability of the proteins (*Figure 4I*). Thus, the combined data led us to conclude that Snf1 activation in glucose-starved cells mediates temporal TORC1 inhibition primarily through phosphorylation of Pib2 at Ser$^{268/309}$, which serves to weaken the association between Pib2 and Kog1 and prevent it from activating TORC1 under these conditions.

## Snf1 constrains Sch9-Thr$^{737}$ phosphorylation by targeting Sch9-Ser$^{288}$

Our phosphoproteomic approach also identified one potential Snf1 target residue, namely Ser$^{288}$, that lies within the C2 domain of Sch9 (*Figure 5A and B*, and *Supplementary file 2C, F*). Like for Pib2, we tried to corroborate these data using in vitro Snf1 kinase assays and an N-terminal fragment of Sch9 as substrate, which, unlike full-length Sch9, could be stably expressed in and purified from bacteria. Here too, wild-type Snf1, but not the kinase-inactive Snf1$^{T210A}$ mutant, was able to phosphorylate this Sch9 fragment, and mutation of Ser$^{288}$ to alanine in the Sch9 fragment significantly reduced the respective phosphorylation level, indicating that this residue is likely the primary Snf1 target in the N-terminal part of Sch9 (*Figure 5—figure supplement 1*). To further study the dynamics of Ser$^{288}$ phosphorylation in vivo, we raised antibodies that specifically recognize phosphorylated Ser$^{288}$ in Sch9 (Sch9-pSer$^{288}$). Using these antibodies, we observed that the Sch9-pSer$^{288}$ levels were barely detectable in *snf1$^{as}$* cells growing exponentially on 2% glucose, but then rapidly and strongly increased in a Snf1-dependent manner following glucose starvation in DMSO-treated cells but not in 2NM-PP1 treated, Snf1-inhibited cells (*Figure 5D*; upper panels, left side). In addition, the Sch9-pSer$^{288}$ levels were quite elevated in cells growing exponentially on very low glucose levels (i.e. 0.05%), but rapidly declined in cells where Snf1$^{as}$ was inactivated by 2NM-PP1 treatment (*Figure 5D*; upper panels, right

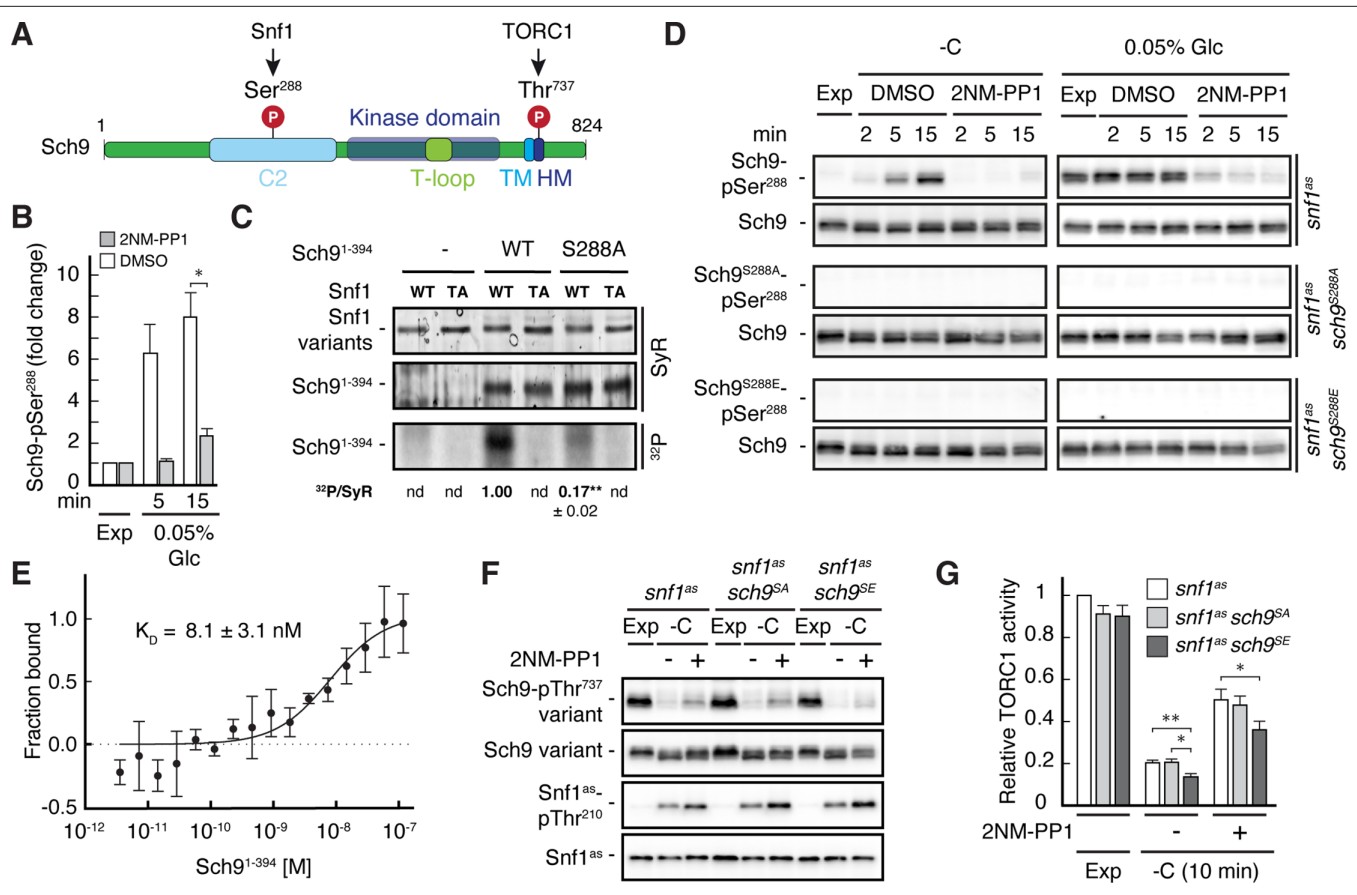

**Figure 5.** Snf1 phosphorylates Sch9-Ser[288] to antagonize Sch9-Thr[737] phosphorylation. (**A**) Schematic representation of the structure of Sch9 with the C2 domain, the kinase domain including the T-loop, the turn motif, and the hydrophobic motif (HM). The position of the potential Snf1 target residue (*i.e.* phosphorylated [P] Ser[288]) and one of the 5 C-terminal TORC1 target residues (***Urban et al., 2007***; i.e. phosphorylated [P] Thr[737] in the HM motif that is used to probe TORC1 activity here) are indicated. (**B**) Phosphoproteomic analyses identify Sch9-Ser[288] as a potential Snf1 target. Mean phosphorylation levels of Sch9-Ser[288] in untreated (DMSO) and 2NM-PP1-treated *snf1as* cells, grown exponentially (Exp) and limited for glucose (0.05% Glc) for the times indicated, were extracted from the SILAC experiment (***Supplementary file 2*** and ***Figure 3***) and normalized to the ones in exponentially growing cells (set to 1.0; n=3; + SD; unpaired Student's t-test, *FDR ≤0.05). (**C**) Snf1 phosphorylates Ser[288] in Sch9 in vitro. Snf1 (WT) and kinase-inactive Snf1[T210A] (TA) were purified from yeast and used in protein kinase assays with [γ-[32]P]-ATP and the N-terminal fragment of Sch9 (encompassing the N-terminal 394 amino acids; Sch9[1-394]) as substrate (WT). In parallel protein kinase assays, we also used the respective Sch9[1-394] fragment harboring the Ser[288]-to-Ala mutation as substrate (S288A). Substrate phosphorylation was detected by autoradiography ([32]P, lower panel) and Sypro Ruby (SyR) staining is shown as loading control for the Snf1 variants (upper panel) and the Sch9 fragments (panel in the middle). The mean phosphorylation of the Sch9[S288A] mutant fragment by wild-type Snf1 (i.e. [32]P signal/Sypro Ruby [SyR] substrate input level; nd, not detected) was assessed relative to the one of the Sch9[WT] fragment (set to 1.0, n=3; ± SEM; unpaired Student's t-test, **p≤0.05; nd, not detected). (**D**) Snf1 controls Sch9-Ser[288] phosphorylation in vivo. Cells (*i.e.* *snf1as*, *snf1as sch9[S288A]*, and *snf1as sch9[S288E]*) were grown exponentially on 2% glucose (Exp) and then starved for glucose (-C) in the absence (DMSO) or presence of 2NM-PP1 for the times indicated (panels on the left). In parallel, cells were grown exponentially (Exp) on low glucose media (0.05%; panels on the right) and then treated with vehicle (DMSO) or 2-NMPP1 for the times indicated. The levels of Ser[288] phosphorylation in Sch9 (Sch9-pSer[288]) and total levels of Sch9 were assayed by immunoblot analyses using anti-Sch9-pSer[288] and anti-Sch9 antibodies, respectively (n=3). (**E**) Snf1 binds Sch9. The binding affinity between Sch9[1-394] (purified from yeast) titrated against the Snf1 complex (also purified from yeast and containing the C-terminally GFP-tagged Snf4 γ-subunit) was assessed by microscale thermophoresis. The dissociation constant ($K_D$; 95% profile likelihood = 3.2–19.1 nM; n=3; ± SEM) was calculated using a nonlinear asymmetric sigmoidal regression. (**F, G**) The phosphomimetic Sch9[S288E] allele compromises proper phosphorylation of the C-terminal Thr[737] residue in Sch9. *snf1as*, *snf1as sch9[S288A]* (*sch9[SA]*), and *snf1as sch9[S288E]* (*sch9[SE]*) cells were grown exponentially (Exp) and then starved for 10 min for glucose (-C) in the absence (-; DMSO) or the presence (+) of 2NM-PP1. Immunoblot analyses of Sch9-pThr[737], Sch9, Snf1-pThr[210], and Snf1 were carried out as in ***Figure 1A*** (**F**). The mean relative TORC1 activities in the three strains were quantified as in ***Figure 1B*** and normalized to the values in exponentially growing *snf1as* cells (set to 1.0; n=6; + SEM; unpaired Student's t-test, *p≤0.05, **p≤0.005) (**G**). The online version of this article includes the following source data for ***Figure 5—source data 1***, data for the graph shown in (**B and E**) and quantifications of the blots in (**C**) and for the graph shown in (**G**); ***Figure 5—source data 2***, uncropped blots, gels and autoradiographies shown in (**C, D and F**); ***Figure 5—source data 3***, raw blots, gels and autoradiographies shown in (**C, D and F**).

The online version of this article includes the following source data and figure supplement(s) for figure 5:

*Figure 5 continued on next page*

*Figure 5 continued*

**Source data 1.** Data for the graph shown in B and E and quantifications of the blots in C and for the graph shown in G.

**Source data 2.** Uncropped blots, gels and autoradiographies shown in C, D and F.

**Source data 3.** Raw blots, gels and autoradiographies shown in C, D and F.

**Figure supplement 1.** Snf1 in vitro kinase assays.

**Figure supplement 1—source data 1.** Uncropped blots.

**Figure supplement 1—source data 2.** Raw blots.

side). As expected, in control experiments performed with strains expressing Sch9$^{S288A}$ or Sch9$^{S288E}$ mutant versions, no Sch9-pSer$^{288}$ signal was detected (middle and lower panels). Together with our finding that the Snf1 complex was also able to bind the N-terminal Sch9 fragment in microscale thermophoresis assays (*Figure 5E*), our data therefore indicate that Ser$^{288}$ in Sch9 is a bona fide Snf1 target residue.

To address the physiological role of Ser$^{288}$ phosphorylation in Sch9, we next studied the relative TORC1 activity upon short-term glucose starvation in 2NM-PP1-treated and -untreated *snf1$^{as}$* strains that expressed either wild-type Sch9, the phosphomutant Sch9$^{S288A}$, or the phosphomimetic Sch9$^{S288E}$ using Sch9-pThr$^{737}$ levels as a proxy. Interestingly, expression of Sch9$^{S288E}$, but not expression of wild-type Sch9 or Sch9$^{S288A}$, slightly but significantly suppressed the unscheduled reactivation of TORC1 in 2NM-PP1-treated Snf1$^{as}$-inhibited glucose-starved cells (*Figure 5F and G*). Thus, our data suggest that Snf1-mediated phosphorylation of Ser$^{288}$ negatively impacts on the capacity of TORC1 to phosphorylate the C-terminal Thr$^{737}$ in Sch9.

## Physiological effects of Sch9$^{S288E}$ and Pib2$^{S268E,S309E}$ are additive

Our data so far indicated that Snf1 controls TORC1 signaling both upstream (Pib2) and downstream (Sch9) of TORC1. To address the question of whether these effects are additive concerning the TORC1-controlled phosphorylation of Thr$^{737}$ in Sch9, we studied *snf1$^{as}$* strains in which the phosphomutant and phosphomimetic variants of Sch9 and Pib2 were expressed separately or combined. In line with our data above, the separate expression of Sch9$^{S288E}$ in Snf1$^{as}$-inhibited (2NM-PP1-treated), glucose-starved cells only weakly suppressed the unscheduled reactivation of TORC1 (i.e. relative Sch9-Thr$^{737}$ phosphorylation), while this effect was already strong in case of the separate expression of Pib2$^{S268E,S309E}$ and even stronger when the expression of Sch9$^{S288E}$ and Pib2$^{S268E,S309E}$ were combined (*Figure 6A and B*). In contrast, the separate and combined expression of Sch9$^{S288A}$ and Pib2$^{S268A,S309A}$ showed, as predicted, no significant effect in the same experiment. Unexpectedly, however, the latter combination did not result in transient reactivation of TORC1, like we observed in glucose-starved, Snf1-compromised cells. This may be explained if TORC1 reactivation would rely on specific biophysical properties of the non-phosphorylated serine residues within Sch9 and Pib2 that are not mimicked by respective serine-to-alanine substitutions. Alternatively, Snf1 may employ additional parallel mechanisms (perhaps through phosphorylation of Tco89, Kog1, and/or other factors; see above) to prevent TORC1 reactivation even when Pib2 and Sch9 cannot be appropriately phosphorylated. While such models warrant future studies, our current data still suggest that Snf1-mediated phosphorylation of Pib2 and Sch9 may be both additive and together sufficient to appropriately maintain TORC1 inactive in glucose-starved cells. Corroborating this conclusion, we found the combined expression of Pib2$^{S268E,S309E}$ and Sch9$^{S288E}$, but not their individual expression, nor individual or combined expression of Pib2$^{S268A,S309A}$ and Sch9$^{S288A}$, to significantly reduce the relative Sch9-Thr$^{737}$ phosphorylation when cells were grown exponentially on low-nitrogen-containing media where TORC1 activity is intrinsically low and Snf1 activity somewhat elevated (*Figure 6C and D*). In line with these data, the *snf1$^{as}$ pib2$^{SESE}$ sch9$^{SE}$* strain also exhibited a slightly higher doubling time than the *snf1$^{as}$* strain that was statistically significant on both low-nitrogen-containing media (i.e. 3.28±0.04 h versus 3.06±0.03 h, respectively; n=3; ± SEM; p<0.05) and standard synthetic complete media (i.e. 1.44±0.02 h versus 1.38±0.01 h, respectively; n=3; ± SEM; p<0.05). Finally, cells that combined the expression of Pib2$^{S268E,S309E}$ and Sch9$^{S288E}$ were also more sensitive to sub-inhibitory concentrations of rapamycin than cells expressing only one of these alleles (*Figure 6E*). Thus, Snf1-mediated fine-tuning of TORC1 activity relies on the proper phosphorylation of both Pib2 and Sch9.

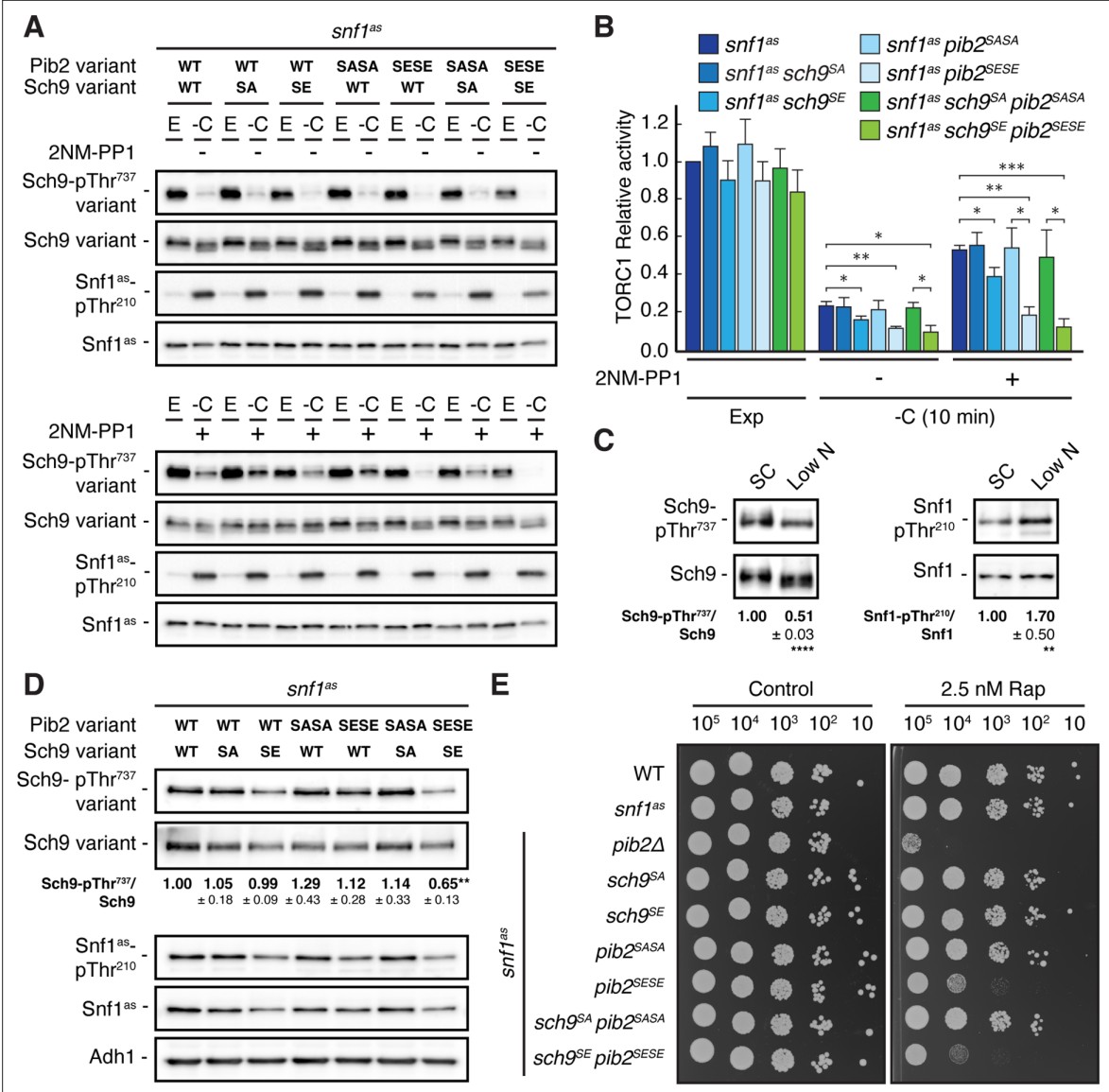

**Figure 6.** Physiological effects of Sch9$^{S288E}$ and Pib2$^{S268E,S309E}$ are additive. (**A, B**) Exponentially growing (E) *snf1*$^{as}$ cells expressing the indicated combinations of wild type Pib2 (Pib2 variant: WT), Pib2$^{S268A,S309A}$ (Pib2 variant: SASA), and Pib2$^{S268E,S309E}$ (Pib2 variant: SESE) and wild type Sch9 (Sch9 variant: WT), Sch9$^{288A}$ (Sch9 variant: SA), and Sch9$^{S288E}$ (Sch9 variant: SE) were starved for glucose (10 min; -C) in the absence (-; DMSO; upper panels) and presence of 2NM-PP1 (+; lower panels). Immunoblot analyses were performed as in *Figure 1A* and the mean relative TORC1 activities (i.e. Sch9-pThr$^{737}$/Sch9) were quantified, normalized relative to exponentially growing *snf1*$^{as}$ cells (set to 1.0), and shown in the bar diagram in (**B**) (n=4; + SEM; unpaired Student's t-test, *p≤0.05, **p≤0.005, ***p≤0.0005). (**C**) Growth on media containing low nitrogen levels activates Snf1-Thr$^{210}$ phosphorylation and decreases TORC1 activity. WT cells were grown exponentially on SC or low nitrogen medium and assayed for their mean relative TORC1 activities (Sch9-pThr$^{737}$/Sch9) and Snf1-Thr$^{210}$ phosphorylation levels (Snf1-pThr$^{210}$/Snf1), each normalized to the values of cells grown on SC (set to 1.0; n=10; ± SEM). In unpaired Student's tests, both values in cells growing on low nitrogen medium were significantly different form the ones growing on SC (**p≤0.005, ****p≤0.005). (**D, E**) Combined expression of Sch9$^{S288E}$ and Pib2$^{S268E,S309E}$ causes reduced Sch9-Thr$^{737}$ phosphorylation (**D**) and enhanced rapamycin sensitivity (**E**). In (**D**), indicated strains were grown exponentially on low nitrogen medium and assayed for their mean TORC1 activities as in (**C**), which were normalized to the value of WT cells (set to 1.0; n=4; ± SEM; unpaired Student's t-test, **p≤0.005). In (**E**), cells with the indicated genotypes were grown exponentially in SD, then 10-fold serially diluted, spotted on control plates (SD) or 2.5 nM rapamycin-containing plates, and grown for 3 days at 30 °C (n=4). The online version of this article includes the following source data for *Figure 6—source data 1*, quantifications of the blots in (**C and D**) and for graph shown in (**B**); *Figure 6—source data 2*, uncropped blots shown in (**A, C and D**); uncropped dropspots shown in (**E**); *Figure 6—source data 3*, raw blots shown in (**A, C and D**); raw dropspots shown in (**E**).

The online version of this article includes the following source data for figure 6:

**Source data 1.** Quantifications of the blots in C and D and for graph shown in B.

*Figure 6 continued on next page*

*Figure 6 continued*

**Source data 2.** Uncropped blots shown in A, C and D; uncropped dropspots shown in E.

**Source data 3.** Raw blots shown in A and C.

**Source data 4.** Raw blots shown in D; raw dropspots shown in E.

## Discussion

Snf1/AMPK and TORC1 are conserved central kinase modules of two opposing signaling pathways that control cell growth and metabolism in response to the availability of nutrients and energy. Accordingly, energy depletion activates Snf1/AMPK, which helps to maintain energy homeostasis by favoring catabolic and inhibiting anabolic processes that generate and consume ATP, respectively (*Hardie et al., 2012*). In contrast, intracellular nutrients and high energy levels activate TORC1, which favors anabolic processes such as lipid, nucleotide, and protein synthesis and inhibits the catabolic autophagic recycling of macromolecules (*Albert and Hall, 2015*; *González and Hall, 2017*; *Laplante and Sabatini, 2012*; *Liu and Sabatini, 2020*). Given the reciprocal cellular roles of Snf1/AMPK and TORC1, it is not surprising that their signaling pathways are wired to each other at different levels to coordinate the establishment of cellular homeostasis. TORC1, for instance, phosphorylates and thereby inhibits the catalytic subunit of AMPK in both mammals and the fission yeast *Schizosaccharomyces pombe* (*Ling et al., 2020*). Conversely, AMPK inhibits mTORC1 through phosphorylation of TSC2 and Raptor (*Gwinn et al., 2008*; *Inoki et al., 2003b*), while Snf1 contributes to TORC1-body formation through phosphorylation of Kog1, although this does not seem to be required for rapid TORC1 inactivation, nor for maintaining TORC1 inactive following acute glucose starvation (*Hughes Hallett et al., 2015*; *Sullivan et al., 2019*). In this context, the heterodimeric Gtr1-Gtr2 Rag GTPase complex has been mechanistically linked to the rapid glucose-starvation induced inactivation of TORC1 and its assembly into oligomeric structures coined TOROIDs (TORC1 organized in inhibited domains) (*Prouteau et al., 2017*), but these processes occur independently of Snf1 and rely on still elusive mechanisms that link glucose signals to the nucleotide-loading state of the Rag GTPases. Despite these findings, Snf1 may be more intimately linked to TORC1 than current knowledge suggests, because both our current and previous proteomics approaches identified several additional potential Snf1 target residues within TORC1 and some of its upstream regulators (*Braun et al., 2014*; *Kanshin et al., 2017*). Here, we followed up on one of these leads, that is Pib2, and demonstrated that the Snf1-mediated phosphorylation of Pib2 is critical to maintain TORC1 inactive during the early phase of acute glucose starvation. Thus, our data extend the function of Pib2 to a hub that integrates both glutamine, as reported earlier (*Tanigawa and Maeda, 2017*; *Tanigawa et al., 2021*; *Ukai et al., 2018*), and glucose signals to control TORC1. Our current data favor a model according to which Snf1-mediated phosphorylation of the Kog1-binding domain in Pib2 weakens its affinity to Kog1 and thereby reduces the TORC1-activating influence of Pib2 that is mediated by the C-terminal TORC1-activating (CAD) domain via a mechanism that is still largely elusive. Interestingly, Pib2 has also been involved in the formation of TORC1-bodies, which seem not to be required for TORC1 inactivation per se but rather serve to increase the threshold for TORC1 reactivation after long-term glucose starvation (*Sullivan et al., 2019*). The latter process is primarily driven by phosphorylation of Kog1 and, albeit dependent on Pib2, may also require loosening of the Kog1-Pib2 association, because deletion of the N-terminal TORC1-inhibitory domain (NID) in Pib2 has been found to induce TORC1-body formation (*Sullivan et al., 2019*). Here too, the mechanistic details remain to be deciphered and will likely require detailed structural information on the interactions between Pib2, Kog1, and the EGOC that also plays a role in TORC1-body formation (*Sullivan et al., 2019*). Curiously, human phafin/LAPF, which is structurally most closely related to Pib2 and with which it shares the FYVE and CAD domains (*Kim and Cunningham, 2015*), also regulates mTORC1-controlled processes, but it remains currently unknown whether these effects are executed through regulation of mTORC1 (*Hatakeyama, 2021*).

The activities of the Snf1 and TORC1 pathways are also wired to each other through a set of directly or indirectly controlled common effectors that, given the opposite roles of Snf1 and TORC1 in cell metabolism, are generally inversely regulated by these pathways (*De Virgilio, 2012*). For instance, Snf1 and TORC1 converge on the transcription factor Gln3 and the eukaryotic translation initiation factor 2α (eIF2α) to oppositely regulate their function (*Beck and Hall, 1999*; *Bertram et al., 2002*; *Cherkasova et al., 2010*; *Cherkasova and Hinnebusch, 2003*). Interestingly, Snf1 and TORC1 have

already previously been reported to directly converge on different residues within Sch9 (*Lu et al., 2011*), a key controller of protein synthesis and aging in yeast (*Loewith and Hall, 2011*). Accordingly, both kinases phosphorylate Sch9, but, while the TORC1 target residues in Sch9 have been functionally well-defined and located in the C-terminus of Sch9 (*Urban et al., 2007*), the respective Snf1 target residue(s) remained unidentified (*Lu et al., 2011*). Here, we show that Snf1 phosphorylates Ser[288] in Sch9, which is not only supported by our current and previous phosphoproteomic approaches (*Braun et al., 2014*), but also by our extended biochemical in vitro and in vivo studies of wild-type and Sch9[S288A] alleles. In line with the opposite conceptual roles of Snf1 and TORC1 within cells, our physiological experiments posit a model in which Snf1-mediated phosphorylation of Ser[288] in Sch9 antagonizes the TORC1-mediated phosphorylation (at Thr[737]) and hence activation of Sch9. Notably, such a model also provides an elegant explanation for the reciprocal role of Snf1 and Sch9 in controlling chronological life span (*Maqani et al., 2018*; *Wei et al., 2008*; *Wierman et al., 2017*), although it is difficult to reconcile with the idea that Snf1 activates Sch9 (in parallel to TORC1) to inhibit replicative lifespan (*Lu et al., 2011*). The latter assumption, however, appears to depend in part on the phosphorylation of a threonine(s) in Sch9, specifically in cells lacking the regulatory Snf1 subunit Sip2, and may therefore not be related to the Sch9-Ser[288] phosphorylation studied here. Of note, the question of how Ser[288] phosphorylation restrains TORC1-mediated phosphorylation of Sch9-Thr[737] also warrants further experimental efforts. The Ser[288] residue lies within the C2 domain that has been suggested to mediate the vacuolar recruitment of Sch9 (*Jin et al., 2014*). However, recent studies have elaborated that the membrane-localization of Sch9 is primarily defined by its N-terminal 184 amino acids, but not the C2 domain (*Chen et al., 2021*; *Novarina et al., 2021*). We therefore assume that the phosphorylation of Ser[288] within the C2 domain, rather than controlling the subcellular localization, may either favor the recruitment of a pThr[737]-targeting phosphatase to Sch9 or interfere with appropriate TORC1 docking. The latter could for instance be achieved if the phosphorylated C2 domain would act as an auto-inhibitory domain that folds back to the C-terminal part of Sch9 to impose specific conformational constraints. Intriguingly, a similar regulatory mechanism has been suggested for the protein kinase Ypk1, a TORC2 target that is very closely related to Sch9 (*Thorner, 2022*). In this case, structural predictions suggested that a critical aspartate (D242) located in a N-terminal C2-like domain is packed tightly against the upper lobe of the kinase domain in Ypk1 to favor its inactive state (*van Dam et al., 2011*). For Sch9, as for Pib2, the resolution of these issues will likely require more detailed structural information on Sch9 that is currently not available.

In this study, we focused on the mechanisms through which Snf1 regulates TORC1 during the early phase of acute glucose starvation. Our phosphoproteome data, however, indicate that Snf1 may also control TORC1 through processes that are relevant during later phases of prolonged glucose starvation. For instance, we identified the lipid kinase Fab1, which generates phosphatidylinositol-3,5-bisphosphate (PI(3,5)$P_2$) from PI3P, as a potential Snf1 target (*Figure 3F*). Fab1 is not only a TORC1 effector, but it also defines the activity and subcellular distribution (between vacuolar membranes and signaling endosomes) of TORC1 (*Chen et al., 2021*). Snf1 may therefore modulate the feedback control loop between Fab1 and TORC1 in response to glucose limitation. This function is likely executed by Snf1 complexes containing the β-subunit Sip1, which, in addition to being potentially feedback-controlled by Snf1, tethers Snf1 complexes to vacuolar membranes in response to carbon stress (*Hedbacker et al., 2004*). We also identified Snf1-regulated phospho-residues in Apl6, a subunit of the adaptor protein 3 (AP-3) complex that functions in cargo-selective protein transport from the TGN to the vacuole (*Cowles et al., 1997*). In this case, Snf1 may control the flux of Ego1 towards (and hence assembly of the TORC1-regulatory EGOC at) vacuolar membranes, which requires proper AP-3 function (*Hatakeyama et al., 2019*). Our mass spectrometry approaches further highlighted several phospho-residues in α-arrestins including Aly2, Bul1, and Bul2 that appeared to be regulated by Snf1. These arrestins drive Rsp5-mediated ubiquitination of specific nutrient transporters such as the amino acid permeases Gap1, Dip5, and others to orchestrate their endocytosis in response to nutrients (*Kahlhofer et al., 2021*; *Zbieralski and Wawrzycka, 2022*). Because amino acid permeases control TORC1 either via their role in distributing amino acids across cellular membrane compartments or through their proposed role as transceptors (*Melick and Jewell, 2020*; *Wolfson and Sabatini, 2017*; *Zheng et al., 2016*), it is conceivable that Snf1 also controls TORC1 indirectly via phosphorylation of α-arrestins. In a similar vein, Snf1 may also reduce ammonium uptake by impinging on the Npr1-regulated Par32, which inhibits the ammonium transporters Mep1 and Mep3 and thereby indirectly

reduces TORC1 activity (*Boeckstaens et al., 2015*; *Varlakhanova et al., 2018*). Finally, two protein kinases, namely Ptk2 and Hkr1, which activate the plasma membrane $H^+$-ATPase Pma1 in response to glucose (*Eraso et al., 2006*; *Goossens et al., 2000*), were also among the presumed Snf1 targets in our data set (*Figure 3F*). Through Ptk2 and Hkr1, Snf1 may conceivably adjust TORC1 activation that is related to Pma1 and $H^+$-coupled nutrient uptake (*Saliba et al., 2018*). In sum, our proteomics study provides a wealth of new leads for future studies on how Snf1 may be wired to TORC1 to ensure cellular homeostasis under long-term glucose limitation.

## Materials and methods
### Yeast strains, plasmids, and growth conditions
*Saccharomyces cerevisiae* strains and plasmids are listed in *Supplementary file 1A* and *Supplementary file 1B*. Point mutations were introduced in the genome by CRISPR-Cas9, as described (*Generoso et al., 2016*), while gene deletion and tagging were performed using the pFA6a system-based PCR-toolbox (*Janke et al., 2004*). The oligos used to generate the CRISPR-Cas9 plasmids are listed in *Supplementary file 1C*. Plasmids were created as described earlier (*Generoso et al., 2016*). Strains were rendered prototrophic, unless stated otherwise, by transforming them with the empty centromeric plasmids listed in *Supplementary file 1B*. In order to maintain the plasmids, cells were pregrown in a synthetic dropout (SD; 0.17% yeast nitrogen base, 0.5% ammonium sulfate [AS], 0.2% dropout mix [USBiological], and 2% glucose) medium. Then, synthetic complete (SC; SD with all amino acids) medium was used for the dilution of the cells the following day. The same procedure was adopted for experiments where cells were grown in media containing low glucose (i.e. 0.05% instead of 2% glucose) or low nitrogen (with a reduced amount of AS [0.0625%] and devoid of amino acids). Starvation experiments were performed by cell filtration and transfer to carbon starvation medium (SC without glucose), nitrogen starvation medium (2% glucose, 0.17% yeast nitrogen base) or nitrogen and carbon starvation medium (0.17% yeast nitrogen base), for the times indicated. For re-addition experiments, 2% final glucose and/or 3.3 mM glutamine was/were added to the culture. When indicated, 10 µM 2NM-PP1 dissolved in DMSO was added to the culture. As a control, the same volume of DMSO was added. Strains and plasmids are available upon request. Cell growth was monitored by measuring the concentration ($OD_{600nm}$/mL) with a spectrophotometer.

### Growth tests on plates
Cells were pregrown over-day in SD or SC until $OD_{600nm}$ above 1.0. Cells were washed two times with water and starting from the concentration of 1.0 $OD_{600nm}$/mL, 10-fold serial dilutions were prepared in water. Cells were spotted on SD or SC plates with or without rapamycin, at the indicated concentrations, and further grown for 3 days at 30 °C.

### SILAC conditions and cell lysis
Yeast strains were grown in SC medium containing either non-labeled or labeled lysine and arginine variants ('medium-heavy' L-arginine-$^{13}C_6$ (Arg6) and L-lysine-$^2H_4$ (Lys4), or 'heavy' L-arginine-$^{13}C_6$-$^{15}N_4$ (Arg10) and L-lysine-$^{13}C_6$-$^{15}N_2$ (Lys8) amino acids (Sigma-Aldrich)), until reaching an $OD_{600nm}$ of approximatively 1.0. Then, cultures grown in the presence of 'medium-heavy' and 'heavy' arginine and lysine were filtered and resuspended in carbon starvation medium for 5 and 15 min, respectively, in the presence of the vehicle (DMSO) or 2NM-PP1. All cultures were ultimately collected by filtration. The nitrocellulose filter was dipped in tubes containing 40 mL of the cell culture medium plus TCA, 6% final concentration. Cells were pelleted and washed with 40 mL acetone and subsequently dried overnight in a freeze-dryer (ZIRBUS). Dried differentially labeled cells (30 mg) of each sample were mixed. Cells were lysed in 50 mL tubes with 6 mL urea buffer (8 M urea, 50 mM Tris HCl pH 8.0) and acid-washed glass beads using a Precellys machine (6x30 s, with 60 s pause after each cycle). Debris were pelleted and the supernatants containing cellular proteins were collected, followed by MS sample preparation.

### On beads in vitro kinase assay (OBIKA)
Cell pellets were obtained by five different cultures of exponentially growing *snf1^as* cells. Cells were pre-grown overnight in 5% glucose YP (yeast extract-peptone) and the following day, they were diluted at the concentration of 0.2 $OD_{600nm}$/mL in 2 L 5% glucose YP (yeast extract-peptone). Cells

were grown until late exponential phase, when they were treated with 10 µM 1NM-PP1 for 20 min. Cells were then collected by filtration, frozen in liquid nitrogen, and cryogenically disrupted by using a Precellys homogenizer in 10 mL of primary amine-free lysis buffer (50 mM HEPES pH 7.5, 1% NP-40, 150 mM NaCl, 1 mM EDTA pH 8.0, 1 mM EGTA pH 8.0 and Roche complete protease inhibitor EDTA-free) and acid-washed glass beads using a Precellys machine (6x30 s, with 60 s pause after each cycle). Lysates were collected by centrifugation at 4000 rpm at 4 °C. The lysates were dialyzed using dialysis buffer (50 mM HEPES pH 7.5, 0.1% NP-40, 150 mM NaCl, 1 mM EDTA pH 8.0, 1 mM EGTA pH 8.0, and 1 mM PMSF) and a molecular-porous membrane tubing (14 kDa, Sigma-Aldrich) to remove primary amine containing metabolites. After 2 h at 4 °C, the buffer was refreshed for overnight dialysis. N-hydroxy-succinimide (NHS) -activated Sepharose beads (5 mL) were washed three times with 10 mL of ice-cold 1 mM HCl and two times with 10 mL of lysis buffer before incubating with 60 mg protein to saturate the beads. The coupling was performed on a rotating mixer at 4 °C overnight. Next, the beads were spun down to remove the supernatant. Beads were then washed three times with 10 mL of phosphatase buffer (50 mM HEPES, 100 mM NaCl, 0.1% NP-40). Phosphatase buffer (1 mL) containing 5'000–10'000 units of lambda phosphatase with 1 mM $MnCl_2$ was added and incubated for 4 h at room temperature or overnight at 4 °C on a rotating mixer to dephosphorylate endogenous proteins. Beads were washed two times with 10 mL of kinase buffer (50 mM Tris-HCl pH 7.6, 10 mM $MgCl_2$, 150 mM NaCl, and 1 x PhosSTOP). Endogenous kinases bound to beads were inhibited by incubation with 1 mM FSBA in 1 mL of kinase buffer at RT on the rotor for 2 h. In addition, staurosporine was added to a final concentration of 100 µM to inhibit the remaining active kinases for 1 h. The beads were washed three times with 10 mL of kinase buffer to remove non-bound kinase inhibitors. The supernatant was removed completely by gel loading tips. Kinase buffer was added to a volume of 860 µL for both kinase inactive and wild-type Snf1 samples. Subsequently, 100 µl of 10 mM ATP, 10 µL of 100 mM DTT, and 30 µl of purified kinase variants were added into each tube. Kinase assays were performed on a rotor at 30 °C for 4 h. Finally, reactions were quenched by snap freezing in liquid nitrogen and samples were lyophilized overnight. Urea buffer (250 µL) was added to the dry beads, followed by MS sample preparation (*Hu et al., 2021*).

## MS sample preparation

For in vivo phosphoproteome and OBIKA samples, lysates or proteins on beads were reduced with 1 mM DTT, alkylated with 5 mM iodoacetamide, and digested with endoproteinase Lys-C for 4 h. The concentration of urea was diluted to 1 M before overnight trypsin digestion. The peptides were purified and fractionated as described previously (*Hu et al., 2019*). Briefly, peptides were purified by SPE using HR-X columns in combination with C18 cartridges. The purified peptides were frozen, lyophilized, and fractionated by HpH reversed-phase chromatography (*Batth et al., 2014*). A total of 96 fractions were mixed with an interval of 12 to yield 8 final fractions. The peptides were acidified, frozen in liquid nitrogen, and lyophilized before phosphopeptide enrichment.

For manual phosphopeptide enrichment, samples were incubated with 2 mg $TiO_2$ slurry, which was pre-incubated with 300 mg/mL lactic acid in 80% acetonitrile/1% trifluoroacetic acid (TFA) before enrichment for 30 min at room temperature (*Zarei et al., 2016*). For peptide elution, $TiO_2$ beads were transferred to 200 µL pipette tips, which were blocked by C8 discs. Tips were sequentially washed with 200 µL of 10% acetonitrile/1% TFA, twice with 200 µL of 80% acetonitrile/1% TFA, and 100 µL of LC-MS grade water. Phosphopeptides were eluted into single tubes with 50 µL of 1.25% ammonia in 20% acetonitrile and 50 µL of 1.25% ammonia in 80% acetonitrile. Eluates were acidified with 5 µL of formic acid. Samples were concentrated by vacuum concentration and resuspended in 20 µL of 0.1% formic acid for LC-MS/MS analysis. The tip flow-through was desalted by STAGE tips for non-phosphopeptide analysis.

Automated phosphopeptide enrichment was performed on an Automated Liquid Handling Platform (Bravo, Agilent) (*Post et al., 2017*). The Fe (III)-NTA cartridges (5 µL) were primed with 0.1% TFA in acetonitrile and equilibrated with 0.1% TFA in 80% acetonitrile (equilibration/washing buffer). Peptides were resuspended in 200 µL of equilibration buffer and loaded on the cartridges with a flow rate of 5 µL/min. Cartridges were washed twice with 200 µL of washing buffer with a flow rate of 10 µL/min. Phosphopeptides were eluted with 100 µL of 1% ammonia in 80% acetonitrile with a flow rate of 5 µL/min. Eluates were acidified with 5 µL of formic acid. Samples were concentrated by lyophilizer and resuspended in 20 µL of 0.1% formic acid for LC-MS/MS analysis.

## LC-MS/MS

LC-MS/MS measurements were performed on a QExactive (QE) Plus, HF-X, and Exploris480 mass spectrometer coupled to an EasyLC 1000 and EasyLC 1200 nanoflow-HPLC, respectively (all Thermo Scientific). Peptides were fractionated on a fused silica HPLC-column tip (I.D. 75 µm, New Objective, self-packed with ReproSil-Pur 120 C18-AQ, 1.9 µm (Dr. Maisch) to a length of 20 cm) using a gradient of A (0.1% formic acid in water) and B (0.1% formic acid in 80% acetonitrile in water): samples were loaded with 0% B with a max. pressure of 800 Bar; peptides were separated by 5–30% B within 85 min with a flow rate of 250 nL/min. The spray voltage was set to 2.3 kV and the ion-transfer tube temperature to 250 °C; no sheath and auxiliary gas were used. Mass spectrometers were operated in the data-dependent mode; after each MS scan (mass range m/z=370–1750; resolution: 70'000 for QE Plus and 120'000 for HF-X and Exploris480) a maximum of ten scans for QE Plus, 12 scans HF-X and 20 scans for Exploris480 were performed using a normalized collision energy of 25%, a target value of 10'000 (QE Plus and HF-X)/5000 (Exploris480) and a resolution of 17'500 for QE Plus, 30'000 for HF-X and 60'000 for Exploris480. MS raw files were analyzed using MaxQuant (version 1.6.2.10; *Cox and Mann, 2008*) using a Uniprot full-length *S. cerevisiae* database plus common contaminants such as keratins and enzymes used for in-gel digestion as reference. Carbamidomethylcysteine was set as fixed modification and protein amino-terminal acetylation, serine-, threonine-, and tyrosine- phosphorylation, and oxidation of methionine were set as variable modifications. The MS/MS tolerance was set to 20 ppm and three missed cleavages were allowed using trypsin/P as enzyme specificity. Peptide, site, and protein FDR based on a forward-reverse database were set to 0.01, the minimum peptide length was set to 7, the minimum score for modified peptides was 40, and the minimum number of peptides for identification of proteins was set to one, which must be unique. The 'match-between-run' option was used with a time window of 0.7 min. MaxQuant results were analyzed using Perseus (*Tyanova et al., 2016*).

## MS data analyses

The in vivo phosphoproteome data were analyzed as described (*Hu et al., 2019*). Briefly, measurements of the log2 fold changes on each site were combined into a random effect model, considering a priori the sites as a random effect, and including the variability among replicates by also considering the replicates as a random effect. The model assigns an average effect size and its corresponding 95% confidence interval to each site. If the confidence interval includes values of zeros, then there is no statistically significant log2 fold change, whereas if the confidence interval is above (below) zero, there is statistical evidence for upregulation (downregulation). Additionally, imputation processes were applied on both protein and phosphosite levels. Proteins that were quantified in at least two biological replicates were kept and missing values were replaced by random values of a normal distribution to mimic low abundance measurements. Both width and down shift were applied according to Perseus default settings. Phosphosites were further normalized to the protein levels. Only sites which were quantified in at least three replicates in either DMSO 5 min or 15 min were kept. Missing values in 2NM-PP1-treated samples were then replaced by the maximum likelihood estimators (MLE) imputation method (*Messer and Natarajan, 2008*). Finally, a t-test (FDR≤0.05) was performed between DMSO and 2NM-PP1-treated samples to determine significantly changing phosphosites.

OBIKA data were analyzed using Perseus. Phosphosites which were quantified in at least three replicates in WT samples were kept. Missing values in kinase-inactive samples were replaced by either random values of a normal distribution to mimic low abundance measurements, both width and down shift were applied according to default settings, when none of five replicates was quantified, or MLE, when at least 1 of five replicates was quantified. T-tests (FDR≤0.05) were performed between WT and kinase-inactive samples to identify significantly changing sites.

## Protein purification and in vitro kinase assays

The Snf1 complex was purified from a Snf1-TEV-TAP-expressing yeast strain grown in YPD. To induce Snf1 activation, cells were washed on a filter. The same procedure was adopted to purify the catalytically inactive Snf1 complex containing the *snf1^T210A* α-subunit and the Snf1 complex, containing the C-terminally GFP-tagged Snf4 γ-subunit. Yeast cells carrying the plasmids for Sch9^1-394-TEV-TAP or Sch9-TEV-TAP expression were grown overnight in SRafinose-Ura medium supplemented with 0.01% sucrose. The next day, in order to induce Sch9^1-394-TEV-TAP and Sch9-TEV-TAP expression, 2% (final

concentration) galactose was added to the cells when they reached $OD_{600nm}$/mL of 0.2. The induction with galactose was carried out for 6 h.

For all the protein purification from yeast, cells were then collected by filtration, frozen in liquid nitrogen, and cryogenically disrupted by using a Precellys homogenizer in 10 mL of lysis buffer (50 mM Tris-HCl pH 7.5, 150 mM NaCl, 0.1% NP-40, 10% glycerol, 400 mM Pefabloc, and Roche complete protease inhibitor EDTA-free). The cleared lysates were incubated for 2 h at 4 °C with IgG-coupled Dynabeads (Dynabeads M-270 Epoxy; Invitrogen, Thermo Fisher Scientific, Basel, Switzerland). The beads were washed five times with lysis buffer and proteins were eluted in TEV buffer (50 mM Tris-HCl pH 7.5, 0.5 mM EDTA,) with 2% TEV protease and stored at –80 °C after the addition of 10% glycerol.

$His_6$-Pib2$^{221-635}$ variants were purified from *E. coli* as described in *Péli-Gulli et al., 2015* using Ni-charged agarose beads (QIAGEN, product number 30210). Proteins were eluted in elution buffer (50 mM $NaH_2PO_4$ pH 8.0, and 200 mM imidazole) and stored at –80 °C after the addition of 10% glycerol.

In vitro radioactive kinase reactions were carried out in Snf1 kinase buffer (20 mM HEPES pH 7.5, 100 mM NaCl, 0.5 mM EDTA, 0.5 mM DTT, and 5 mM MgAc), with 60 ng Snf1 (quantified with respect to the Snf1 subunit), 1 µg Pib2, or 80 ng Sch9 in 20 µL total volume, and started by adding the ATP Mix (3 µL [γ–32P]-ATP [Hartmann Analytic, SRP-501], 6 µL 200 µM ATP, and 1 µL Kinase Buffer 1 X) and stopped by adding SDS-PAGE sample buffer. Reactions were carried out at 30 °C and for 10 min or 30 min for Pib2 or Sch9, respectively. Proteins were separated by SDS-PAGE, stained with SYPRO Ruby (Sigma) to assess loading, and analysed using a phosphoimager (Typhoon FLA 9500; GE Healthcare). In vitro kinase assays probed by immunoblot analysis was carried out similarly as described above. The reaction was performed in 40 µL volume for 30 min at 30 °C. In the ATP mix, [γ-$^{32}$P]-ATP was substituted with $H_2O$. Finally, the reaction was probed using the following antibodies: custom-made rabbit anti-Sch9-pSer$^{288}$ (Eurogentec, 1:4000), and goat anti-Sch9 (GenScript, 1:1000). To assess the loading, the gel was stained with SYPRO Ruby (Sigma).

## Co-immunoprecipitation

Yeast cells expressing the indicated fusion proteins were harvested by filtration, frozen in liquid nitrogen, and cryogenically disrupted using the Precellys homogenizer in 4 mL of lysis buffer (50 mM Tris-HCl pH 7.5, 150 mM NaCl, 0.1% NP-40, 10% glycerol, 400 mM Pefabloc, and Roche complete protease inhibitor EDTA-free) with the addition of 60 mM glutamine as in *Ukai et al., 2018*. Cleared lysates were equilibrated in the same lysis buffer. For input samples, aliquots of cleared lysates were collected and denatured in presence of SDS-PAGE sample buffer. For co-immunoprecipitations, the cleared lysates were incubated for 2 h at 4 °C with prewashed anti-c-myc MagBeads (Pierce Thermo Fisher Scientific, product number 88843). After five washes with lysis buffer, beads were resuspended in 20 µL lysis buffer and denatured in presence of SDS-PAGE sample buffer. Inputs and pull-down samples were analyzed by SDS-PAGE immunoblot with mouse anti-myc (Santa Cruz, 1:10,000) and mouse anti-HA (ENZO, 1:1000) antibodies.

## Microscale thermophoresis

Microscale thermophoresis (MST) experiments were performed using a Monolith NT.115 (Nanotemper Technologies). Labeled purified Snf1 complex (0.144 µM), containing the C-terminally GFP-tagged Snf4 γ-subunit, was mixed with a twofold serial dilution (a total of 16 samples) of unlabeled 18.1 µM $His_6$-Pib2$^{221-635}$ in elution buffer (50 mM $NaH_2PO_4$ pH 8.0, 200 mM imidazole, and 10% glycerol) or with unlabeled 0.118 µM Sch9$^{1-394}$ in elution buffer (50 mM Tris-HCl pH 8.0, 0.5 mM EDTA, and 10% glycerol). Samples were loaded into Monolith NT.115 Capillaries and MST measurements were performed using 20% laser power setting at 30 °C. Experiments were performed in triplicates and data were fitted using the $K_d$ model of the MO.Affinity Analysis software (Nanotemper Technologies). The dissociation constant $K_d$ was obtained by plotting the bound fraction against the logarithm of ligand concentration.

## Cell lysate preparation and immunoblot analyses

Cell lysates were prepared similarly as described in *Hatakeyama et al., 2019*. After denaturation at 98 °C for 5 min, samples were loaded on SDS-PAGE and transferred onto nitrocellulose membranes. After 1 h blocking with blocking buffer (5% milk powder in tris-buffered saline), membranes were

immunoblotted with the following primary antibodies: rabbit anti-Adh1 (Calbiochem, product number 126745; 1:200,000 dilution), rabbit anti-Sch9-pThr$^{737}$ (De Virgilio lab, 1:10,000 dilution), rabbit anti-Sch9-pSer$^{288}$ (Rospert lab, 1:4000 dilution), goat anti-Sch9 (De Virgilio lab, 1:1000 dilution), rabbit anti-AMPK-pThr$^{172}$ (Cell Signal, product number 2535 S, 1:1000 dilution) to detect the phosphorylation of Snf1-Thr$^{210}$, mouse anti-His$_6$ (Sigma, product number H1029, 1:1000 dilution) to detect total levels of Snf1, anti-ACC1-pSer$^{79}$ (Thermo Fisher Scientific, product number PA5-17564, 1:500 dilution), and mouse anti-GFP (Roche, product number 11814460001, 1:3000 dilution). After 3 washes, the membranes were incubated with anti-mouse (BIO-RAD, product number 170–6516; 1:3000 dilution), rabbit (BIO-RAD, product number 170–6515; 1:3000 dilution), or goat (BIO-RAD, product number 5160–2104; 1:3000 dilution) secondary antibodies conjugated with horseradish peroxidase, washed again for three times, and developed with ECL (GE Healthcare).

## Statistical analyses

Statistical significance was determined by three or more independent biological replicates, by using Student's t-test analysis, performed with GraphPad Prism 9.0. Unpaired Student's t-test was used for the comparison of normalized data. Values with a p-value (or FDR where indicated) lower than 0.05 were considered significantly different. In the figure legends, the number of independent replicas, method used to express the variability, specific statistical tests, and significance are indicated.

*Figures 2–6*

## Acknowledgements

We thank Riko Hatakeyama and Dieter Kressler for plasmids, Malika Jaquenoud for technical assistance, Marie-Pierre Péli-Gulli for carefully reading the manuscript and guidance with microscopy, Roberto Pagliarin for the kind gift of the 2NM-PP1 inhibitor, Sebastien Leon and Andrew Capaldi for constructive criticism based on the preprint version of this article, and Jeremy Thorner for pointing out the mechanistic aspects of Ypk1 regulation. This work was supported by the Canton of Fribourg (to J D and C D V), the Swiss National Science Foundation (310030_166474/184671 to CDV, 310030_184781 to JD, and 316030_177088 to JD and CDV), FWO-Vlaanderen (G069413, G0C7222N) and KU Leuven (C14/17/063, C14/21/095) to JW. We also acknowledge financial support from the Italian Ministry of University and Research (MUR) through grants 2020-ATE-0329 and Dipartimenti di Eccellenza 2017 to the University of Milano-Bicocca, Department of Biotechnology and Biosciences (to FT and PC) and from the Deutsche Forschungsgemeinschaft (DFG, German Research Foundation) through Project-ID 403222702 - SFB 1381, TP B08 (to SR) and RO 1028/5–2 (to SR).

## Additional information

### Funding

| Funder | Grant reference number | Author |
| --- | --- | --- |
| Canton of Fribourg | | Jörn Dengjel<br>Claudio De Virgilio |
| Schweizerischer Nationalfonds zur Förderung der Wissenschaftlichen Forschung | 310030_166474/184671 | Claudio De Virgilio |
| Schweizerischer Nationalfonds zur Förderung der Wissenschaftlichen Forschung | 310030_184781 | Jörn Dengjel |

| Funder | Grant reference number | Author |
|---|---|---|
| Schweizerischer Nationalfonds zur Förderung der Wissenschaftlichen Forschung | 316030_177088 | Jörn Dengjel Claudio De Virgilio |
| Fonds Wetenschappelijk Onderzoek | G069413 | Joris Winderickx |
| Fonds Wetenschappelijk Onderzoek | G0C7222N | Joris Winderickx |
| Katholieke Universiteit Leuven | C14/17/063 | Joris Winderickx |
| Katholieke Universiteit Leuven | C14/21/095 | Joris Winderickx |
| Ministero dell'Università e della Ricerca | 2020-ATE-0329 | Farida Tripodi Paola Coccetti |
| Deutsche Forschungsgemeinschaft | Project-ID 403222702 - SFB 1381 | Sabine Rospert |
| Deutsche Forschungsgemeinschaft | RO 1028/5-2 | Sabine Rospert |
| Deutsche Forschungsgemeinschaft | TP B08 | Sabine Rospert |

The funders had no role in study design, data collection and interpretation, or the decision to submit the work for publication.

## Author contributions

Marco Caligaris, Conceptualization, Formal analysis, Validation, Investigation, Visualization, Methodology, Writing – review and editing; Raffaele Nicastro, Conceptualization, Formal analysis, Supervision, Validation, Investigation, Visualization, Methodology, Project administration, Writing – review and editing; Zehan Hu, Conceptualization, Data curation, Formal analysis, Validation, Investigation, Visualization, Methodology, Writing – review and editing; Farida Tripodi, Joris Winderickx, Sabine Rospert, Paola Coccetti, Conceptualization, Validation, Writing – review and editing; Johannes Erwin Hummel, Validation, Investigation, Methodology, Writing – review and editing; Benjamin Pillet, Conceptualization, Formal analysis; Marie-Anne Deprez, Validation, Writing – review and editing; Jörn Dengjel, Conceptualization, Resources, Formal analysis, Validation, Methodology, Writing - original draft, Writing – review and editing; Claudio De Virgilio, Conceptualization, Resources, Data curation, Supervision, Funding acquisition, Validation, Methodology, Writing - original draft, Writing – review and editing

## Author ORCIDs

Marco Caligaris (ID) http://orcid.org/0000-0003-1732-7694
Raffaele Nicastro (ID) http://orcid.org/0000-0002-5420-2228
Farida Tripodi (ID) http://orcid.org/0000-0003-1246-979X
Benjamin Pillet (ID) http://orcid.org/0000-0002-7313-4304
Joris Winderickx (ID) http://orcid.org/0000-0003-3133-7733
Sabine Rospert (ID) http://orcid.org/0000-0002-3089-9614
Paola Coccetti (ID) http://orcid.org/0000-0001-5898-5883
Jörn Dengjel (ID) http://orcid.org/0000-0002-9453-4614
Claudio De Virgilio (ID) http://orcid.org/0000-0001-8826-4323

## Decision letter and Author response

Decision letter https://doi.org/10.7554/eLife.84319.sa1
Author response https://doi.org/10.7554/eLife.84319.sa2

## Additional files

### Supplementary files
- Supplementary file 1. Strains, plasmids and oligonucleotides used in this study.
- Supplementary file 2. Significantly in vivo and in vitro regulated phosphosites.
- MDAR checklist

### Data availability

All data generated or analyzed during this study are included in the manuscript and supporting files. Source data files have been provided for Figures 1-6 and figure supplements. The mass spectrometry proteomics data have been deposited to the ProteomeXchange Consortium via the PRIDE partner repository with the dataset identifier PXD037381.

The following dataset was generated:

| Author(s) | Year | Dataset title | Dataset URL | Database and Identifier |
|---|---|---|---|---|
| Dengjel J | 2022 | Snf1/AMPK fine-tunes TORC1 signaling in response to glucose starvation | https://www.ebi.ac.uk/pride/archive/projects/PXD037381 | PRIDE, PXD037381 |

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
