## [Editor Report]

This rigorous and careful study provides some of the first mechanistic insights into the way that glucose starvation triggers inhibition of TORC1 (particularly in yeast) and will serve as an important resource for those interested in AMPK/Snf1 dependent regulation of a variety of other pathways and processes. The paper also provides the clearest picture yet of the regulation of Pib2, an important but poorly understood TORC1 regulator in yeast and likely beyond. The proposed mechanism is interesting and proposes multiple ways of interaction between the two signaling cascades, and will be of interest to researchers working on mechanisms of gene regulation by signaling pathways.

---

## [Decision Letter]

**Decision letter after peer review:**

Thank you for submitting your article "Snf1/AMPK fine-tunes TORC1 signaling in response to glucose starvation" for consideration by *eLife*. Your article has been reviewed by 3 peer reviewers, and the evaluation has been overseen by a Reviewing Editor and Kevin Struhl as the Senior Editor. The reviewers have opted to remain anonymous.

Essential revisions:

Although no new experiments are required, there are several issues to address, and the authors should consider all the comments in the reviews (which may overlap).

1) Please comment on the apparent inconsistency of the SE and SA mutants. The authors could tone down their claims by saying that yet another layer of regulation might contribute to TORC1 silencing in the absence of Snf1 phosphorylating Pib2 and Sch9. Also, the phosphorylation of Pib2 is supposedly inhibitory, so the SE allele should be equivalent to the loss of function (indeed, the effect of the SE allele is the strongest argument that phosphorylation is inhibitory). However, the SA allele should be gain-of-function, but this doesn't seem to be the case; ie, resistant to glucose starvation.

2) Please comment on the subtlety of the impact of Snf1 on TORC1, particularly the possibility of redundant mechanisms. Perhaps there are overlapping layers of TORC1 regulation in glucose starvation (e.g. TORC1-bodies/TOROIDS and regulation of Gtr1/2).

3) Please present representative immunoblots and ideally verifying key observations with an additional robust assay (such as cleaved Sch9).

4) Please discuss the phosphoproteomic data, particularly the list of Snf1 sites, and analyze potential motifs.

*Reviewer #2 (Recommendations for the authors):*

The authors limit their analysis with PIB2 to glucose-limited conditions. Given that this protein was identified as a glutamine-responsive regulator of TORC1, it would be appropriate to address how glutamine impacts the role of the SNF1-dependent phosphorylation of PIB2. Here the question is whether SNF1 is responsible for only glucose-mediated changes or whether glutamine and glucose converge at the same positions phosphorylated by SNF1.

*Reviewer #3 (Recommendations for the authors):*

To strengthen the observations of Snf1-mediated suppression of TORC1 activity via Pib2, the authors may investigate other TORC1 read-outs. Also, authors can use the Sch9 mobility shift assay as shown by Hughes Hallet et al. to validate their observations.

The physiological relevance of the proposed mechanisms by which Snf1 represses TORC1 might be investigated by analyzing the growth under low glucose media conditions.

The kinase assay in figures 4D and 5C does not show Snf1 levels. Probing for the Snf1 WT and TA will solidify the observation. Also, why purified Pib2 migrate in two bands in figure 4D?

---

## [Author Response]

Essential revisions:Although no new experiments are required, there are several issues to address, and the authors should consider all the comments in the reviews (which may overlap).1) Please comment on the apparent inconsistency of the SE and SA mutants. The authors could tone down their claims by saying that yet another layer of regulation might contribute to TORC1 silencing in the absence of Snf1 phosphorylating Pib2 and Sch9. Also, the phosphorylation of Pib2 is supposedly inhibitory, so the SE allele should be equivalent to the loss of function (indeed, the effect of the SE allele is the strongest argument that phosphorylation is inhibitory). However, the SA allele should be gain-of-function, but this doesn't seem to be the case; ie, resistant to glucose starvation.

We fully agree with this comment and now address it in the text, as outlined in more details in our response to point 2 of reviewer #1 below.

2) Please comment on the subtlety of the impact of Snf1 on TORC1, particularly the possibility of redundant mechanisms. Perhaps there are overlapping layers of TORC1 regulation in glucose starvation (e.g. TORC1-bodies/TOROIDS and regulation of Gtr1/2).

We have, throughout the entire text, rephrased the description of the role of Snf1 in TORC1 control, which is limited to a period of transient reactivation of TORC1 during

the early phase of glucose starvation. In addition, we now also discuss the Snf1-independent role of the Rag GTPases (i.e. Gtr1 and Gtr2) in mediating TORC1 inactivation and TOROID formation during acute glucose starvation.

3) Please present representative immunoblots and ideally verifying key observations with an additional robust assay (such as cleaved Sch9).

We have replaced the respective immunoblots (in each case an issue was raised; Figure 1A, 1B, and 5F). We have also added input controls for Snf1 in Figures 4D and 5C as requested. Regarding the NTCB-cleaved Sch9 assay, please see our answer to point 9 of reviewer #3.

4) Please discuss the phosphoproteomic data, particularly the list of Snf1 sites, and analyze potential motifs.

We have now analyzed, as suggested by reviewer #1 (point 1), the potential difference between early (5 min) and late (15min) responding Snf1-sites. This new analysis yielded indeed some interesting additional insight that we now discuss appropriately in the Results section (for more details please see our specific response to point 1 of reviewer #1 below).

Reviewer #2 (Recommendations for the authors):The authors limit their analysis with PIB2 to glucose-limited conditions. Given that this protein was identified as a glutamine-responsive regulator of TORC1, it would be appropriate to address how glutamine impacts the role of the SNF1-dependent phosphorylation of PIB2. Here the question is whether SNF1 is responsible for only glucose-mediated changes or whether glutamine and glucose converge at the same positions phosphorylated by SNF1.

To address the question to what extent nitrogen (including glutamine) limitation activates Snf1 when compared to glucose starvation, we subjected wild-type cells to nitrogen or glucose starvation and measured Snf1 activity by determining the levels of Snf1-Thr^210^ and ACC1-GFP-Ser*^79^* phosphorylation (new Supplementary figure 1B and 1C). While glucose starvation strongly activated Snf1, the respective activation by nitrogen starvation was, albeit significant, rather moderate. Accordingly, the Snf1 activity in nitrogen-starved cells was approximately 10 times lower than the one in glucose-starved cells. Hence, Snf1 mediates to some extent nitrogen starvation signals, but glucose starvation is by far more important in this context (which is why we primarily focus on this condition here).

Reviewer #3 (Recommendations for the authors):To strengthen the observations of Snf1-mediated suppression of TORC1 activity via Pib2, the authors may investigate other TORC1 read-outs. Also, authors can use the Sch9 mobility shift assay as shown by Hughes Hallet et al. to validate their observations.

Using the anti Sch9-pThr^737^ antibodies to assess TORC1 activity is in our view still the best currently available TORC1 readout. We have also used the proposed band-shift assays (of the NTBC-cleaved Sch9 C-terminus) in the past but realized that the quantification of TORC1 activity is less accurate. Accordingly, the Sch9 C-terminus migrates in several phospho-isoforms, not all of which rely on direct TORC1-mediated phosphorylation. For instance, Thr^570^ is phosphorylated by Pkh1/2 (PMID: 17560372). In addition, Ser^723^ and Ser^726^ (formerly considered to be direct TORC1 sites; PMID: 17560372), have recently been shown to be targets of the CDK Bur1 (PMID: 35166010). Our own current studies show in fact that the latter two residues define the turn motif of Sch9 and are also targeted by the CDK Pho85-Pho80, which primes Sch9 for subsequent phosphorylation of its Thr^737^ residue (similarly as CDC2 and GSK-3 do in mammalian S6K; PMID: 9271440, 11914378, 12586835, and 22065737). The responsiveness of these phospho-residues to rapamycin is most likely due to the activation of a phosphatase. Thus, we hope it is acceptable to use Sch9-pThr^737^/Sch9 levels as a TORC1 readout here.

The physiological relevance of the proposed mechanisms by which Snf1 represses TORC1 might be investigated by analyzing the growth under low glucose media conditions.

Please see our response to point 4 of reviewer #2.

The kinase assay in figures 4D and 5C does not show Snf1 levels. Probing for the Snf1 WT and TA will solidify the observation. Also, why purified Pib2 migrate in two bands in figure 4D?

We included the input levels of Snf1 in the revised Figures 4D and 5C. Notably, Pib2 gets partially degraded during the preparation, which is why there are additional degradation bands visible. We modified Figure 4D accordingly and highlighted this issue in the legend.